# Low levels of tetracyclines select for a mutation that prevents the evolution of high-level resistance to tigecycline

**Jennifer Jagdmann, Dan I. Andersson, Hervé Nicoloff** [ID] *

Uppsala University, Department of Medical Biochemistry and Microbiology, Uppsala, Sweden

* herve.nicoloff@imbim.uu.se

## Abstract

In a collection of *Escherichia coli* isolates, we discovered a new mechanism leading to frequent and high-level tigecycline resistance involving tandem gene amplifications of an efflux pump encoded by the *tet*(A) determinant. Some isolates, despite carrying a functional *tet*(A), could not evolve high-level tigecycline resistance by amplification due to the presence of a deletion in the TetR(A) repressor. This mutation impaired induction of *tetA*(A) (encoding the TetA(A) efflux pump) in presence of tetracyclines, with the strongest effect observed for tigecycline, subsequently preventing the development of *tet*(A) amplification-dependent high-level tigecycline resistance. We found that this mutated *tet*(A) determinant was common among *tet*(A)-carrying *E. coli* isolates and analysed possible explanations for this high frequency. First, while the mutated *tet*(A) was found in several ST-groups, we found evidence of clonal spread among ST131 isolates, which increases its frequency within *E. coli* databases. Second, evolution and competition experiments revealed that the mutation in *tetR*(A) could be positively selected over the wild-type allele at sub-inhibitory concentrations of tetracyclines. Our work demonstrates how low concentrations of tetracyclines, such as those found in contaminated environments, can enrich and select for a mutation that generates an evolutionary dead-end that precludes the evolution towards high-level, clinically relevant tigecycline resistance.

**Data Availability Statement:** All relevant data is available within the manuscript and Supporting Information files.

## Introduction

Species such as *Escherichia coli* have a large, open pan-genome, and a substantial portion of the genome of any individual isolate consists of accessory genes [1], allowing for extensive variation in both gene content and sequence between isolates [2]. This genetic variation can substantially impact the ability of individual isolates to develop new phenotypes such as antibiotic resistance. For example, the presence of tandem-repeated sequences can favour spontaneous genetic amplifications that may increase resistance [3]. Thus, to obtain a more realistic picture of the potential for resistance development, it is important to examine a set of isolates that better encompasses the full diversity of a bacterial species' pan-genome.

**Funding:** This work was supported by a grant to DIA from the Swedish Research Council https://www.vr.se (2017-01527) and Wallenberg Foundation https://kaw.wallenberg.org (KAW 2018.0168). The funders had no role in study design, data collection and analysis, decision to publish, or preparation of the manuscript.

**Competing interests:** The authors have declared that no competing interests exist.

**Abbreviations:** CFU, colony-forming unit; CSLI, Clinical and Laboratory Standards Institute; CT, chlortetracycline; DMSO, dimethyl sulfoxide; DO, doxycycline; EUCAST, European Committee on Antimicrobial Susceptibility Testing; MI, minocycline; MIC, minimum inhibitory concentration; MHA, Mueller–Hinton agar; MHB, Mueller–Hinton broth; OT, oxytetracycline; ST, sequence type; TC, tetracycline; TGC, tigecycline; WGS, whole-genome sequencing.

Examining the impact of genetic variability of the pan-genome on resistance evolution may be especially relevant for antibiotics where our understanding of resistance is incomplete, such as for the last-resort antibiotic tigecycline (TGC). TGC is a tetracycline (TC) antibiotic, and tetracyclines have been widely used for human therapy, animal husbandry, and agriculture [4]. Tetracyclines are linear, tetracyclic molecules with functional groups that vary between the different antibiotics [4–6]. Several common tetracycline resistance determinants in gram-negative bacteria are efflux pumps such as the class A tetracycline efflux pump encoded on the *tet*(A) determinant, consisting of the *tetA*(A) and *tetR*(A) genes coding for the pump and its transcriptional regulator, respectively [7–10]. Expression of *tetA*(A) is regulated by the transcriptional repressor TetR(A) that binds as a dimer to the operator site of *tetA*(A) and blocks transcription [11]. When tetracycline is present, it binds to TetR(A), resulting in release of the repressor and expression of the TetA(A) efflux pump.

The increase in TC resistance led to the development of second-generation tetracycline antibiotics, such as doxycycline (DO) and minocycline (MI) in the 1970s, followed by the third-generation antibiotic TGC (from the glycylcycline class) in the 2000s [12–16]. TGC has improved binding to the ribosome while maintaining enhanced activity against bacteria carrying various tetracycline resistance determinants such as *tet*(A) [17,18]. In spite of these improvements, clinical failure of TGC treatment in Enterobacteriales is still observed [19–21]. However, the mutations isolated in *E. coli* under laboratory selection (e.g., in *acrR*, *marR*, *rpsJ*, lipopolysaccharide-associated genes, or point mutations in *tet*(A)) only confer a moderate reduction in susceptibility (TGC MICs at or below 0.5 mg/L, unless in combination [up to 0.75 mg/L] or overexpressed in the case of mutated *tet*(A)), indicating that our current understanding of the clinically relevant TGC resistome is incomplete [22–28]. This discrepancy between observed high TGC MICs appearing in isolates during antimicrobial treatment (e.g., 8 [19] and 24 [20] mg/L; clinical breakpoint >0.5 mg/L according to EUCAST) and failure to evolve high-level TGC resistance when using laboratory strains likely emphasises the impact that intra-species diversity has on the ability of each individual isolate to develop high-level resistance.

In this work, we investigated the potential of *E. coli* to develop high-level TGC resistance by analysing a collection of isolates in an approach that better takes the genetic diversity of this species into account. We showed that amplifications of *tet*(A) can lead to high-level TGC resistance. Unexpectedly, a substantial fraction of the examined isolates was unable to develop TGC resistance by amplification of *tet*(A) due to a mutation in *tetR*(A) that reduces induction of *tet*(A). We show that this *tetR*(A) mutant variant is associated with the successful ST131 clone and is selected at low (sub minimal inhibitory concentration, sub-MIC) antibiotic concentrations, showing (counter-intuitively) that weak antibiotic pressure can select for a mutation preventing subsequent evolution of high-level resistance. This is in contrast to the common observation that several mutations, each conferring a small increase in resistance, accumulate under sub-MIC selection in a stepwise manner to generate high-level resistance [29].

## Results

### Identification of TGC resistance mechanisms

To identify potential new mechanisms of spontaneous TGC resistance, 16 TGC-susceptible and phylogenetically diverse *E. coli* isolates (see S1 Table) were screened for their ability to develop resistance when exposed to 0.5, 1, and 2 mg/L TGC (TGC clinical breakpoint ≤0.5 mg/L, according to EUCAST, January 1, 2019). One isolate carrying *tet*(A) (DA44554) developed resistance on TGC 0.5 mg/L plates at a high frequency ($2.82 \times 10^{-1}$), and unlike the other

isolates, also allowed selection of mutants on plates supplemented with 1 and 2 mg/L TGC. To further study the potential role of $tet$(A) in TGC resistance development, more isolates with $tet$(A) were added to the screen. In this process, a frequent allele of $tet$(A) with a 24-bp deletion in $tetR$(A) (hereafter referred to as $tet$(A)$^{\Delta tetR}$) was identified in 16 isolates that were included in the screen (Fig 1A). In total, an additional 53 isolates with $tet$(A) (37 with $tet$(A)$^{wt}$ and 16 with $tet$(A)$^{\Delta tetR}$) and 3 isolates without $tet$(A) were screened. The majority (34 out of 38) of the isolates carrying $tet$(A)$^{wt}$ developed TGC resistance at a high frequency when exposed to 0.5 mg/L TGC (median frequency of all isolates carrying $tet$(A)$^{wt}$ $6.24 \times 10^{-4}$), while isolates carrying $tet$(A)$^{\Delta tetR}$ or lacking $tet$(A) developed resistance at much lower frequencies (median frequencies $2.53 \times 10^{-7}$ and $1.66 \times 10^{-8}$, respectively; Fig 1B and S2 Table). Furthermore, first-step mutants of isolates carrying $tet$(A)$^{wt}$ were often selected on plates supplemented with 1 mg/L (35 out of 38 isolates) and/or 2 mg/L TGC (17 out of 38 isolates) but less frequently with isolates carrying $tet$(A)$^{\Delta tetR}$ or lacking $tet$(A) (mutants selected on 1 mg/L TGC for 8 out of 16 and 1 out of 18 isolates, respectively, and on 2 mg/L TGC for only 2 $tet$(A)$^{\Delta tetR}$ isolates and no isolates lacking $tet$(A); S1 Fig and S3 Table). TGC-resistant mutants of isolates carrying $tet$(A)$^{wt}$ and selected on TGC 0.5, 1, and 2 mg/L had MICs above the EUCAST clinical

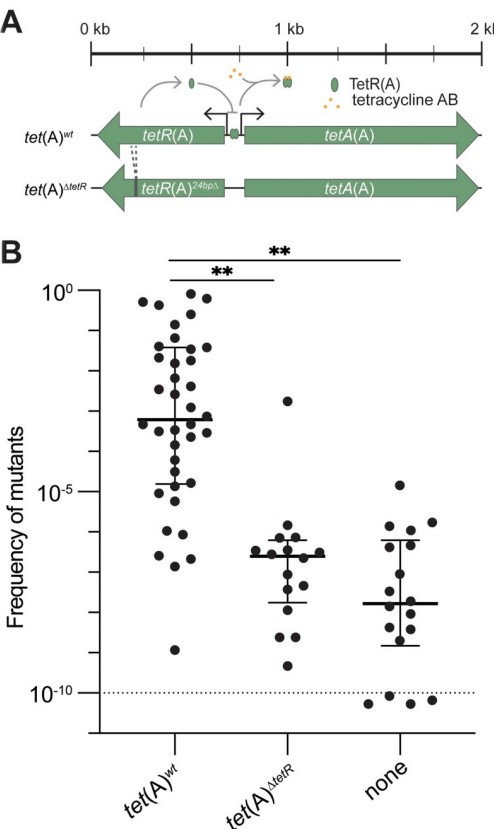

**Fig 1. Relevant $tet$(A) alleles and frequencies of resistant mutants.** (**A**) The $tet$(A)$^{wt}$ and $tet$(A)$^{\Delta tetR}$ variants of $tet$(A), including a schematic of the regulation of $tetA$(A) by TetR(A). (**B**) Frequency of mutants determined by a simplified fluctuation assay at 0.5 mg/L TGC varies depending on $tet$(A) allele. Data points below dashed line (representing limit of detection) are not measured values but represent values below our limit of detection (about $1 \times 10^{-10}$) for determining the mutant frequency of the isolate. Bars indicate median values and error bars represent the interquartile ranges. Isolates carrying $tet$(A)$^{wt}$ had a significantly higher frequency of TGC-resistant mutants than isolates carrying $tet$(A)$^{\Delta tetR}$ or no $tet$(A) (Wilcoxon rank sum test, $p < 0.0001$ (**)). The underlying data can be found in S1 Data. TGC, tigecycline.

breakpoint (MICs reaching up to 1.5 mg/L) in a single selection step (S2 and S3 Tables). Further stepwise selection for higher resistance using a subset of isolates carrying $tet(A)^{wt}$ led to mutants with MICs up to 16 mg/L (32 times above the EUCAST clinical breakpoint, see S1 Method and S1 Fig and S4 Table). Importantly, 15 out of 16 isolates carrying $tet(A)^{\Delta tetR}$ did not develop resistance on TGC 0.5 mg/L at a higher frequency than isolates lacking $tet(A)$ (Fig 1B and S2 Table).

Whole-genome sequencing (WGS) of first-step mutants selected either on TGC 0.5 mg/L (18 mutants) or TGC 1 or 2 mg/L (11 mutants) showed that in isolates carrying $tet(A)^{wt}$, the frequent first-step mutations were spontaneous tandem amplifications that included $tet(A)^{wt}$ (S4 and S5 Tables). Amplifications of up to 15 to 20 copies per plasmid (up to 156× per chromosome) varied in size (approximately 6 to 140 kb in length) and were enabled by directly repeated regions of 400 bp to 1800 bp, often consisting of transposases or transposon-associated elements present on each side of the amplified units (Fig 2 and S4 and S5 Tables). In some of the whole-genome sequenced mutants, an increase in plasmid copy number also contributed to the overall increased gene copy number when $tet(A)^{wt}$ was carried on a plasmid (S4 and S5 Tables). Overexpression of $tetA(A)$ cloned on an expression plasmid confirmed that increased dosage of $tet(A)^{wt}$ was the cause of the increased TGC MIC in the mutants carrying $tet(A)^{wt}$ amplifications and that increased expression of $tetA(A)$ confers a fitness cost (S2A and S2B Fig and S2 Method). Additionally, the increase in mRNA expression with increased DNA copy-number of $tet(A)^{wt}$ was confirmed (S2C Fig). WGS of stepwise-selected mutants of isolates carrying $tet(A)^{wt}$ selected at higher TGC concentrations showed accumulation of additional mutations consisting of point mutations in chromosomal genes (S4 Table and S1 Fig). These point mutations varied between isolates but mostly targeted known genes of the TGC resistome that are involved in low-level resistance [22,25]. A different pattern of resistance development was observed with isolates carrying $tet(A)^{\Delta tetR}$ and isolates lacking $tet(A)$, including those carrying the class B tetracycline efflux pump $tet(B)$ (S2, S3 and S5 Tables). In those isolates, the frequency of resistant mutants on plates supplemented with TGC 0.5 mg/L was low and the MIC of the mutants was often lower than that of mutants isolated with strains carrying $tet(A)^{wt}$ (S2 Table). WGS of a subset of mutants of isolates carrying

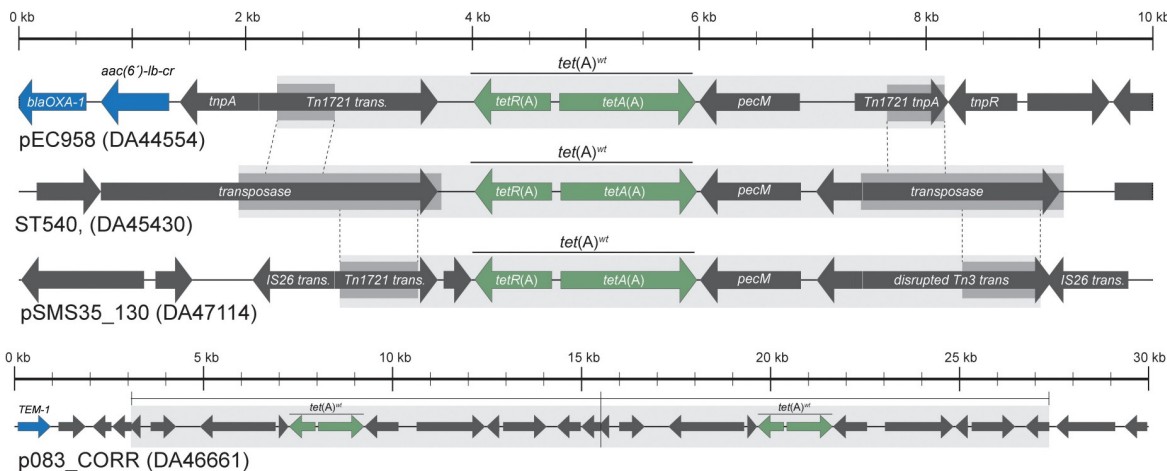

**Fig 2. Genetic context of $tet(A)^{wt}$ and amplified units.** Depicted are the amplified regions carrying $tet(A)^{wt}$ in mutants of DA44554, DA45430, DA47114, and DA46661, selected on 1 mg/L TGC. Genes shown in green are the primary genes of interest ($tetA(A)$ and $tetR(A)$). Dark grey boxes show the regions of direct repeat, while the light grey boxes show the amplified regions. Dashed lines indicate where regions of direct repeat are identical to those found in the other isolates. Genes shown in blue are other resistance genes. For DA46661, an initial duplication was observed in the parental isolate. TGC, tigecycline.

$tet(A)^{\Delta tetR}$ or lacking $tet(A)$ selected at 0.5 mg/L TGC revealed that their increased MICs mostly resulted from point mutations targeting known genes of the TGC resistome and causing low levels of resistance (0.19 to 0.38 mg/L; S5 Table).

Selections performed with a *Klebsiella pneumoniae* isolate carrying $tet(A)^{wt}$ (DA32094) revealed that TGC resistance linked to $tet(A)^{wt}$ amplifications (quantified by qPCR) also occurs in pathogens other than *E. coli* (S3A Fig). This result fits with previous findings of tandem genetic amplifications causing antibiotic resistance in *K. pneumoniae* [3,30]. Additionally, omadacycline, a novel oral tetracycline antibiotic with a structure similar to TGC, was recently approved by the FDA. As seen in S3B and S3C Fig, omadacycline-resistant *E. coli* mutants had increased $tet(A)^{wt}$ copy numbers and overexpression of the $tet(A)^{wt}$ allele led to omadacycline resistance, similar to what was seen for TGC. Thus, carriage and amplification of $tet(A)^{wt}$ appeared to be an early key step in rapid development of TGC and omadacycline resistance.

## A $tet(A)^{\Delta tetR}$ allele with a mutated $tetR(A)$ prevents TGC resistance from developing by gene amplification

While TGC MICs for isolates carrying $tet(A)^{wt}$ were sometimes higher (1 or 2 Etest increments, within the range of error for an Etest, S2 and S6 Tables) than MICs for $tet(A)^{\Delta tetR}$-carrying isolates or isolates without $tet(A)$, the overlap between the MICs for these 2 groups of isolates indicated that a slightly elevated MIC for isolates carrying $tet(A)^{wt}$ alone could not explain the frequent resistance development observed in strains carrying $tet(A)^{wt}$. A key question is why the functional $tet(A)^{\Delta tetR}$ (as indicated by the MICs of isolates carrying $tet(A)^{\Delta tetR}$ for other tetracyclines, S6 Table) could not develop high TGC resistance by $tet(A)^{\Delta tetR}$ amplification. To confirm that the 24-bp deletion in $tetR(A)$ present in the $tet(A)^{\Delta tetR}$ determinant indeed prevents TGC resistance development, the deletion was reconstructed in *E. coli* MG1655 carrying $tet(A)^{wt}$ on plasmid pEC958. Unlike the strain carrying $tet(A)^{wt}$, the isogenic strain carrying $tet(A)^{\Delta tetR}$ had low frequencies of TGC-resistant mutants and did not amplify $tet(A)^{\Delta tetR}$ (S7 Table). Thus, the 24-bp deletion in $tetR(A)$ prevented resistance from evolving by gene amplification. This was further confirmed using mutants of an isolate carrying a $tet(A)^{\Delta tetR}$ allele (DA33135) that had been selected in presence of an aminoglycoside (mutants DA35498 and DA35553 [3]). In those mutants, tobramycin resistance developed by amplification at 11 copies of a 27.7 kb region carrying an aminoglycoside resistance gene *aac(3)-IId*. The amplified unit also co-amplified a $tet(A)^{\Delta tetR}$ determinant, which was also present at 11 copies. However, when tested these mutants and the parental strain (DA33135) had similar MICs for TGC (S4A Fig). Similar results were observed for amplified $tet(A)^{\Delta tetR}$ and omadacycline (S4B Fig). When induced in presence of TGC, we did not observe an increase in $tetA(A)$ mRNA levels in a mutant carrying increased copy numbers of $tet(A)^{\Delta tetR}$ (S2 Fig). These observations indicated that the inability of isolates carrying $tet(A)^{\Delta tetR}$ to develop high TGC resistance was not due to $tet(A)^{\Delta tetR}$ preventing amplification of $tet(A)$ but rather was caused by the 24-bp mutation in $tetR(A)$ preventing the gene dosage increase of $tet(A)^{\Delta tetR}$ from increasing TGC MIC (S4 Fig).

## Occurrence of the $tet(A)^{\Delta tetR}$ allele, $tet(A)$ diversity, and genetic context

Since the $tet(A)^{\Delta tetR}$ allele was found in several isolates, we examined how common this allele is among *E. coli* isolates and whether other alleles of $tet(A)$ exist. *E. coli* carrying $tet(A)$ were identified in 3 collections of isolates: (i) 200 sequenced *E. coli* with $tet(A)$ were found in the NCBI complete sequenced *E. coli* database; (ii) 38 isolates with $tet(A)$ were identified in a collection of 333 sequenced uropathogenic (UPEC) and blood *E. coli* isolates [31]; and (iii) 73 isolates with $tet(A)$ were identified by PCR and local sequencing among 504 bacteraemia *E. coli*

isolates from our own collection of strains. In summary, 10.5% (NCBI), 36.8% (UPEC and blood isolates), and 23.3% (in-house bacteraemia collection) of the isolates carried the 24-bp deletion in $tetR(A)$ found in the $tet(A)^{\Delta tetR}$ allele (Fig 3A). Less common, smaller deletions and truncations of $tetR(A)$ were also identified (Figs 3A and 3B and S5), and 2.7% to 17.5% of the isolates carried a 28-bp truncation of the 3′ end of $tetA(A)$ resulting in a TetA(A) protein with 9 amino acids at the C-terminal end altered (Figs 3A and 3C and S5). However, this mutation had no effect on TetA(A) activity (as measured by MIC, S6 Table) and did not prevent TGC resistance from developing by $tet(A)$ amplification (see strain DA63164 in S2 Table). Overall, the analysis revealed a previously undescribed diversity of $tet(A)$ (see Figs 3 and S5) and, most importantly, that the $tet(A)^{\Delta tetR}$ allele occurred frequently.

The sequence type (ST) group was determined for all $tet(A)$-carrying *E. coli* isolates from NCBI where a complete genome was available (S6 Fig and S8 Table) and had previously been determined in the UPEC and blood isolate study [31]. When applicable, plasmid type for isolates from NCBI carrying $tet(A)$ on a plasmid was determined (S6 and S7 Figs and S8 Table). The probability of an isolate carrying the $tet(A)^{\Delta tetR}$ allele to be an ST131 clone was higher

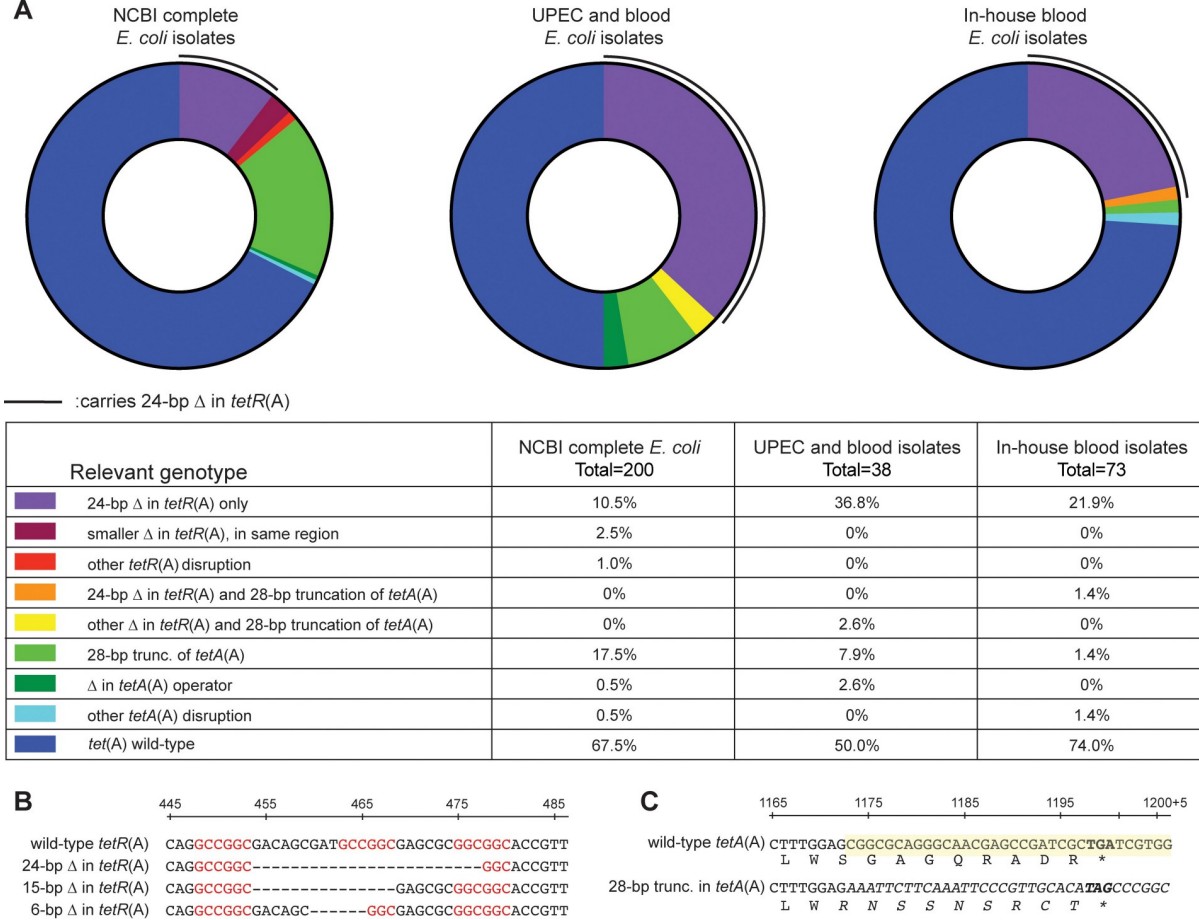

**Fig 3. Occurrence of *tet*(A) alleles.** (**A**) Occurrence of *tet*(A) alleles in 3 collections of *E. coli*. Black lines represent alleles containing the 24-bp deletion in *tetR*(A). Legend to the figures and detailed frequencies are given in the table. The underlying data can be found in S1 Data. (**B**) Sequence of the region of *tetR*(A) prone to deletions, with regions of high similarity marked in red. (**C**) Sequence and translation of the 28-bp truncation of *tetA*(A). The 28 base pairs at the 3′-end of *tetA*(A) that differ in the *tetA*(A) variant with the 28-bp deletion are highlighted in yellow in the wild-type sequence and italicised in the variant. The resulting alternative amino acids sequence in the variant TetA(A) is italicised. In-frame stop codons are in bold.

than both the probability of any isolate in the UPEC and blood isolate study and the probability of $tet$(A)-carrying *E. coli* isolates from NCBI to belong to ST131 (S1 Results and S6 Fig and S8 Table), indicating clonal spread of $tet$(A)$^{\Delta tetR}$. In ST131 isolates from the $tet$(A)-carrying *E. coli* isolates from NCBI, we found that $tet$(A)$^{wt}$ and $tet$(A)$^{\Delta tetR}$ determinants were associated with different plasmid alleles, indicating that the spread of $tet$(A)$^{\Delta tetR}$ in ST131 isolates might be facilitated by its association with specific plasmids (S1 Results and S7 Fig and S8 Table). Comparing the local genetic context around $tet$(A) for the $tet$(A)-carrying *E. coli* isolates from NCBI revealed that $tet$(A)$^{\Delta tetR}$ was often present in a different genetic context than $tet$(A)$^{wt}$ (S7 Fig; $n = 68$ $tet$(A) analysed, $tet$(A) alleles carrying any other mutation than the 24-bp deletion in $tetR$(A) were excluded). However, we also observed that neither $tet$(A)$^{wt}$ nor $tet$(A)$^{\Delta tetR}$ could be fully tied to a single successful ST group or plasmid, and both alleles were observed in other ST groups than ST131 in both collections (S6 and S7 Figs and S8 Table).

Spontaneous amplifications rely on the presence of large identical repeats present in the same orientation and on each side of the amplified region [32]. To estimate the potential of the $tet$(A)$^{wt}$-allele present in numerous isolates to amplify, we searched for such identical repeats within a 7 kb region on each side of 68 $tet$(A)$^{wt}$ and $tet$(A)$^{\Delta tetR}$ determinants from the isolates carrying $tet$(A) from NCBI (S7 Fig). One or more sets of identical repeats with a size between 458 and 1,786 bp were detected on each side of 31 out of 51 $tet$(A)$^{wt}$. Importantly, repeats involved in $tet$(A)$^{wt}$ amplifications in 3 *E. coli* isolates for which we selected spontaneous TGC-resistant mutants (see Fig 2) were detected around 27 out of 51 $tet$(A)$^{wt}$ alleles, including 8 out of 9 $tet$(A)$^{wt}$ present in ST131 isolates (S7 Fig). Overall, our results show a high potential for $tet$(A)$^{wt}$ amplification and TGC resistance development.

## PCR screen to identify if an isolate can develop clinical TGC resistance

To detect the $tet$(A)$^{wt}$ allele in *E. coli*, we developed a simple and efficient screen that specifically detects and distinguishes between $tet$(A)$^{wt}$ and $tet$(A)$^{\Delta tetR}$ alleles (see S2 Results and S8 Fig). This screen correctly identified all $tet$(A) alleles tested.

## Effect of the 24-bp deletion in *tetR*(A) on repressor function

A persisting question is how could the 24-bp deletion in $tetR$(A) prevent amplification of $tet$(A)$^{\Delta tetR}$ and TGC resistance from developing, while still causing high levels of resistance to other tetracyclines (S6 Table)? Our observation that an increase in $tet$(A)$^{\Delta tetR}$ copy number did not lead to an increase in TGC resistance suggests that the mutated repressor is not responding properly to the inducer (TGC). This could, for example, result from either a reduced affinity of the repressor to tetracycline antibiotics, or an impaired release of the repressor bound to the $tetA$(A) promoter.

To elucidate the impact of the 24-bp deletion in $tetR$(A) on TetA(A) expression, $tetA$(A) mRNA transcript levels were quantified in isogenic $tet$(A)$^{wt}$ and $tet$(A)$^{\Delta tetR}$ strains grown in absence of antibiotics and at 1/10th the $tet$(A)$^{wt}$ MIC of TC, TGC, DO, and MI (Fig 4A and 4B). In presence of TGC and MI, $tetA$(A) expression from $tet$(A)$^{\Delta tetR}$ was reduced compared to $tet$(A)$^{wt}$, whereas for TC and DO, no difference was discernible at the concentration used. Basal induction in absence of tetracyclines was identical for the $tet$(A)$^{wt}$ and $tet$(A)$^{\Delta tetR}$ alleles, implying that the two TetR(A) repressor variants were binding to the $tetA$(A) operator with similar affinities.

Since a functional efflux pump TetA(A) will reduce the intracellular concentration of tetracyclines when induced and will therefore affect induction of $tetA$(A) transcription, as has been described for $tetA$(B) [33], we decoupled the activity of TetR(A) from that of TetA(A) by constructing strains with $tetA$(A)-$lacZ$ translational fusions regulated by either allele of $tetR$(A) and lacking a functional TetA(A) pump. This allowed us to better compare the effect of the

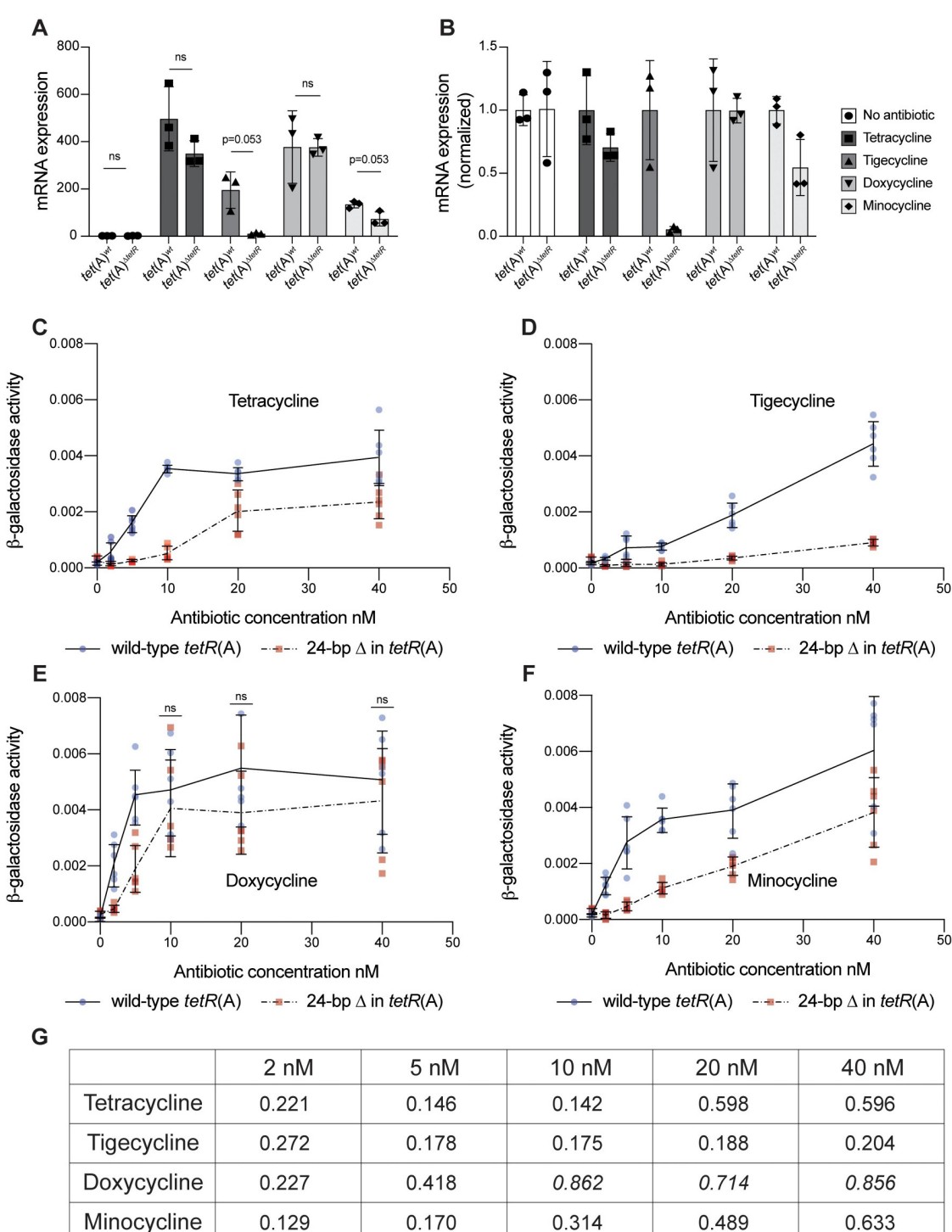

**Fig 4. Expression and induction of *tetA*(A) in response to tetracycline antibiotics.** (**A, B**) mRNA expression levels of *tetA*(A) for
*tet*(A)^wt^ and *tet*(A)^ΔtetR^ (**A**) and mRNA expression levels normalised to expression of *tet*(A)^wt^ in response to induction by different
tetracycline antibiotics at concentrations 1/10th the MIC of the strain carrying *tet*(A)^wt^ (**B**). The strains used are isogenic and only
differ by the nature of the *tet*(A) allele. Three biological replicates performed for each allele and antibiotic, mean (bars) and SD
shown. *P*-values shown in panel A when *p* < 0.1 (Welch *t* tests). ns; no significant statistical difference. (**C–F**) Expression of TetA(A)
in presence of TC (**C**), TGC (**D**), DO (**E**), and MI (**F**) by β-galactosidase assays (*tetA*(A)-*lacZ* translational fusions). Induction by
wild-type *tetR*(A) (blue circles, solid lines) or *tetR*(A) with 24-bp deletion (red boxes, dashed lines) were compared in otherwise
isogenic strains. Unless stated otherwise (ns; not significant), significant differences in expression were detected between the two
*tetR*(A) alleles (unpaired *t* tests performed for the two alleles at each concentration, with correction for multiple *t* tests using the

Holm–Sidak method, $p < 0.05$). Six biological replicates for each allele and concentration, mean represented by the line and error bars show the SD. (**G**) Ratios of β-galactosidase activity (24-bp deletion in *tetR*(A)/wild-type *tetR*(A)) following induction in presence of increasing antibiotic concentrations. Italicised values were not significantly different between the two *tetR*(A) alleles, while other values were significantly different (unpaired *t* tests performed for the 2 alleles at each concentration, with correction for multiple *t* tests using the Holm–Sidak method, $p < 0.05$). The underlying data for all panels can be found in S1 Data. DO, doxycycline; MI, minocycline; TC, tetracycline; TGC, tigecycline.

various antibiotics by performing beta-galactosidase assays in presence of similar concentrations of each drug. The 24-bp deletion in *tetR*(A) consistently induced the *tetA*(A)-*lacZ* fusion less than wild-type *tetR*(A) for TC, TGC, and MI, while the difference was only significant at the lowest concentrations of DO (Fig 4C–4G). These results were consistent with those obtained in presence of a functional TetA(A) (Fig 4A and 4B) and indicated that the effect of the deletion in TetR(A) was a reduction in induction of *tetA*(A) transcription in presence of tetracyclines. This effect varied with each antibiotic, with the strongest effect observed for TGC. We conclude that the main effect of the 8 amino acid deletion in TetR(A) is not to increase the affinity of the repressor for the operator but rather to either reduce the binding of the different tetracyclines to TetR(A) or to decrease the effect of the binding of the different tetracyclines on the activity of the repressor.

## Effect of the 24-bp deletion in *tetR*(A) on fitness

An important question is if the *tet*(A)$^{\Delta tetR}$ allele can be advantageous when carried in *E. coli* isolates since it occurs frequently, especially in ST131 isolates. Potentially, the mutant allele is beneficial and selected over the *tet*(A)$^{wt}$ allele under certain conditions. For example, under weak induction conditions (at low concentrations of tetracyclines), the cost of expressing TetA(A) (S2B Fig and S2 Method) might outweigh the benefit of the increased resistance, and bacteria carrying the *tet*(A)$^{\Delta tetR}$ allele and exhibiting weaker *tetA*(A) induction might have a higher fitness compared to those with the *tet*(A)$^{wt}$ variant. As a result, bacteria with *tet*(A)$^{\Delta tetR}$ may outcompete those with *tet*(A)$^{wt}$ during growth at low tetracycline levels.

To examine this hypothesis, we measured the relative fitness of the *tet*(A)$^{wt}$ or *tet*(A)$^{\Delta tetR}$ variants in otherwise isogenic bacteria by head-to-head competitions (Fig 5A and 5B). Competitions were performed without antibiotic and in presence of stepwise increases in amounts of TC, TGC, DO, and MI, as well as oxytetracycline (OT) and chlortetracycline (CT), two antibiotics used extensively in agriculture and animal husbandry. The *tet*(A)$^{wt}$ allele had a small but significant fitness advantage without antibiotic (selection coefficient of 0.007, Fig 5C). As expected, at high antibiotic concentrations, *tet*(A)$^{wt}$ also had an advantage, due to the higher expression of TetA(A) and corresponding increase in resistance (Fig 5C, green circles). Most importantly, at a broad range of lower antibiotic concentrations, the *tet*(A)$^{\Delta tetR}$ allele outcompeted *tet*(A)$^{wt}$ for all antibiotics, with maximum selection coefficients up to 0.082 (Fig 5C, yellow circles, and Fig 5D).

While the complete loss of resistance genes can be advantageous, the *tet*(A)$^{\Delta tetR}$ allele carried no detectable cost compared to an isogenic strain lacking *tet*(A) in similar head-to-head competitions (see S9 Fig). Thus, we conclude that under weak selection conditions, bacteria carrying *tet*(A)$^{\Delta tetR}$ can outcompete those carrying *tet*(A)$^{wt}$ and are not outcompeted by susceptible bacteria lacking *tet*(A).

## Spontaneous *tetR*(A) 24-basepair deletion and evolution of *tet*(A) during growth in presence of low concentrations of minocycline

Since strains carrying *tet*(A)$^{\Delta tetR}$ have a growth advantage compared to strains carrying *tet*(A)$^{wt}$ in environments with low concentrations of tetracycline drugs, we tested if we could

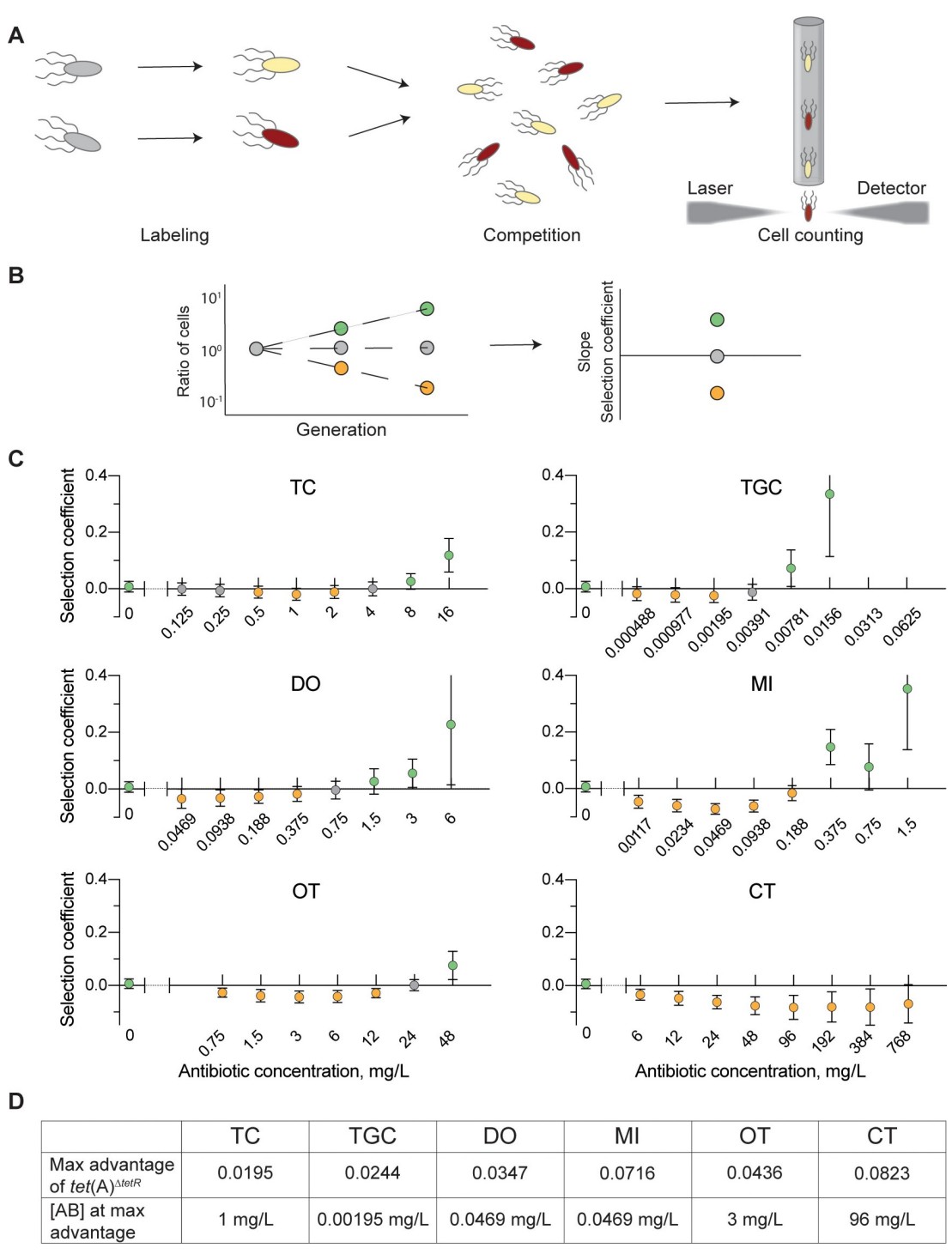

**Fig 5. Competition analysis of strains carrying different *tet*(A) alleles in presence of increasing antibiotic concentrations.**
(**A**) Schematic of head-to-head competition experiments. Isolates to be competed are labelled with fluorescent markers and mixed at equal proportions. After competition growth, cells are counted using a flow cytometer to determine the cell ratio. (**B**) Example of ratios of cells measured over a period incubation (left). The slope of the change in ratio (allele X/allele Y) gives the selection coefficient, which is also plotted on the right. Green: positive selection coefficient (allele X has a fitness advantage over allele Y). Grey: neutral selection (neither allele has a fitness advantage). Orange: negative selection coefficient (allele Y has a fitness advantage over allele X). (**C**) Competitions of isogenic strains carrying either *tet*(A)$^{wt}$ or *tet*(A)$^{\Delta tetR}$, showing the selection coefficient (*tet*(A)$^{wt}$ over *tet*(A)$^{\Delta tetR}$) at different concentrations of TC, TGC, DO, MI, OT, and CT (Wilcoxon rank sum test). X-axis shows the antibiotic concentration in mg/L. The highest concentrations used correspond to 1/2 the MIC of the *tet*(A)$^{wt}$-

carrying strain. Missing values for TGC represent concentrations that were removed as they strongly affected growth of both alleles. Green circles: $tet(A)^{wt}$ has a fitness advantage over $tet(A)^{\Delta tetR}$. Grey circles: no discernible fitness advantage for either allele. Orange circles: $tet(A)^{\Delta tetR}$ has a fitness advantage over $tet(A)^{wt}$. A total of 18 to 25 biological replicates per antibiotic concentration and competition, mean, and SD shown here. The underlying data can be found in S1 Data. (**D**) The maximum fitness advantage of $tet(A)^{\Delta tetR}$, given as the selection coefficient of $tet(A)^{\Delta tetR}$, is indicated as well as the concentration at which the advantage was observed. CT, chlortetracycline; DO, doxycycline; MI, minocycline; OT, oxytetracycline; TC, tetracycline; TGC, tigecycline.

detect occurrences of spontaneous 24-bp deletions in $tetR$(A) and whether those originally low-frequency mutants would enrich in the population when low amounts of tetracyclines are present in the growth medium. In fact, we had noticed that the 24-bp deletion in $tetR$(A) was surrounded by short direct-repeat sequences that might indeed allow for a high frequency of spontaneous deletion (Fig 3B). Based on the previous fitness experiments, lineages of the $tet(A)^{wt}$-carrying isolate used in the competition experiments were cycled for 196 generations with and without antibiotic pressure (0.0469 mg/L MI, a concentration where the fitness advantage of $tet(A)^{\Delta tetR}$ is observed in competition experiments), before screening for presence of $tet(A)^{\Delta tetR}$ at the final time point. Screening by PCR only detected potential $tet(A)^{\Delta tetR}$ in one lineage, so amplicon sequencing of $tetR$(A) was performed on 5 lineages grown with and 10 lineages grown without antibiotic pressure. The 24-bp deletion in $tetR$(A) of the $tet(A)^{\Delta tetR}$ allele was detected in all lineages except one lineage with no antibiotic pressure (Fig 6 and S9 Table). The proportion of $tet(A)^{\Delta tetR}$, when detected, varied between $1.58 \times 10^{-5}$ and $2.06 \times 10^{-1}$ and was generally higher after evolution in presence of MI than without (median frequencies of $1.58 \times 10^{-2}$ and $4.47 \times 10^{-4}$, respectively; Fig 6 and S9 Table). We also noted that, at the end of the evolution experiments, the fluorescence protein expressed in the original isolate was nonfunctional in all or a large proportion of the bacteria in each lineage, suggesting that there is also selection for loss of the fluorescent marker in this experimental set-up.

## Discussion

### TGC resistance evolution by amplification of $tet(A)^{wt}$

In this study, we identified a novel pathway of resistance development to TGC with tandem amplification of $tet(A)^{wt}$ carrying wild-type $tetR$(A) leading to TGC resistance above EUCAST clinical breakpoint in a single selection step (clinical breakpoint ≤0.5 mg/L) [32,34–36]. Furthermore, we showed in a subset of isolates carrying $tet(A)^{wt}$ that following $tet(A)^{wt}$ amplification, accumulation of additional mutations (many in known targets of the TGC resistome) could further increase the MIC above the CLSI clinical breakpoint (clinical breakpoint ≤8 mg/L) [22,25]. Such high MICs have been described for *E. coli* isolates obtained from patients under TGC treatment [20] and were up to 20 times higher than MICs of stepwise-selected mutants of *E. coli* MG1655 [22], revealing the strength of our multi-isolate approach that takes into account the genetic diversity of *E. coli* as a species. We also showed that spontaneous $tet(A)^{wt}$ amplifications led to high frequencies of spontaneous TGC-resistant mutants (on average 3 to 4 log higher frequency than that of isolates carrying $tet(A)^{\Delta tetR}$ or lacking $tet(A)$). In summary, $tet(A)^{wt}$ led to high frequencies of spontaneous clinically relevant TGC-resistant mutants that could further evolve MICs similar to those observed in isolates that developed resistance in patients during antimicrobial treatment. To the best of our knowledge, this is the first description of a common mechanism of evolution towards clinical levels of resistance to TGC in *E. coli*. 89.5% (34/38) of tested $tet(A)^{wt}$-carrying isolates had high frequencies of TGC resistance mutants, corresponding well with the frequent detection of large identical repeats around $tet(A)^{wt}$ (S7 Fig) and that increase rates of spontaneous amplifications [32].

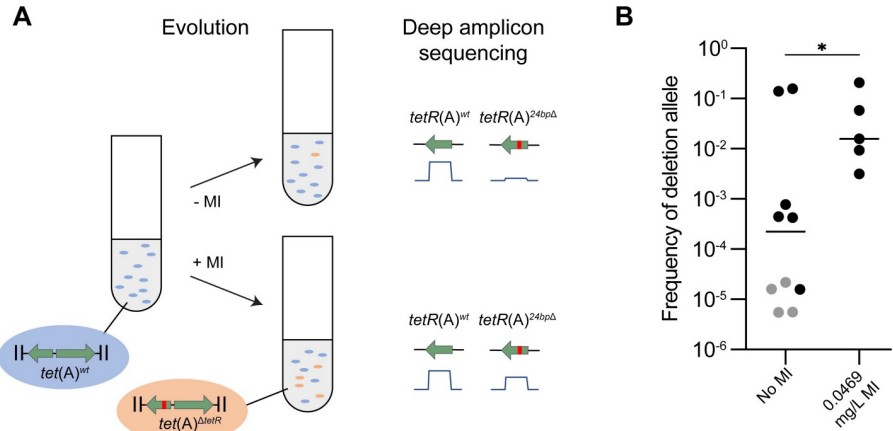

**Fig 6. Evolution of *tet*(A) during prolonged growth.** (**A**) Principle of evolution experiment. A strain carrying a wild-type *tet*(A) with a wild-type *tetR*(A) gene was grown with (+MI) or without (−MI) minocycline. The frequency of bacteria with spontaneous 24-bp deletion in *tetR*(A) (*tet*(A)$^{ΔtetR}$ allele) at the end of the evolution experiment was determined using deep amplicon sequencing and the sequencing coverage for each *tet*(A) allele. (**B**) Frequency of bacteria with *tetR*(A)$^{Δ24bp}$ mutations at the end of the evolution experiment (see S9 Table). Results are after 196 generations without antibiotics ($n = 10$) or 0.0469 ($n = 5$) mg/L MI. In grey are maximum frequencies due to absence of detected *tetR*(A)$^{Δ24bp}$ mutations in at least 1 of the amplicon sequencing technical repeats. Median is shown. *: $p < 0.5$ (Wilcoxon rank sum test). MI, minocycline.

Furthermore, *tet*(A)$^{wt}$ was present on a plasmid in many isolates, indicating that this resistance mechanism is horizontally transferable between isolates and species.

Increased resistance to tetracycline (but not tigecycline) by *tet*(A) amplification has been observed before, and recombinant overexpression of TetA(A) has been shown to decrease TGC susceptibility [28,36]. However, the selection of clinically relevant TGC resistance via *tet*(A)$^{wt}$ amplification has not previously been described, which may be explained by the unstable nature of amplifications [35,37]. For example, 4 strains used in this study (DA21682, DA24516, DA25176, and DA26669 [22]) originally classified as intermediately resistant to TGC when tested in the clinic (by former EUCAST standards, 1 mg/L ≤ MIC < 2 mg/L) were susceptible to TGC (MIC ≤ 0.38 mg/L) when tested in our laboratory. Because these isolates carry *tet*(A)$^{wt}$ and could easily evolve TGC resistance (reaching MICs of 1 mg/L) through *tet*(A)$^{wt}$ amplification, we hypothesise that they might have carried *tet*(A)$^{wt}$ amplifications in the patient and when originally tested in the clinic, but that the amplifications had segregated and the resistance had been lost prior to MIC measurement in our laboratory.

To the best of our knowledge, this is also the first study taking the intra-species genetic diversity into consideration when investigating the ability of a bacterial species to evolve an antibiotic resistance phenotype. Our work reveals the strength of using a pan-genomic approach to investigate new clinically relevant resistance evolution mechanisms that would otherwise remain undiscovered if working solely with a laboratory strain such as *E. coli* MG1655.

## Common mutation in *tetR*(A) impairs TGC resistance evolution by amplification

Our approach also led to the identification of a frequent allele of *tetR*(A) with a 24-bp deletion that prevents TGC resistance from evolving by *tet*(A) amplification. We found that 10.5% to 36.8% of *E. coli* isolates carrying *tet*(A) had the *tet*(A)$^{ΔtetR}$ allele with the in-frame 24-bp deletion in *tetR*(A). An important question is, why do deletions in the transcriptional regulator

TetR(A) prevent TGC resistance from evolving, especially since the deletion in *tetR*(A) has no significant impact on the MICs to tetracyclines? Expression analysis revealed that the 24-bp deletion in *tetR*(A) reduced the ability of all tested tetracyclines to induce the TetA(A) efflux pump to different degrees. This reduced induction by TC, DO, and MI had no significant impact on the MICs to those drugs. However, induction was more reduced with TGC. We showed that even if amplified to 15 copies per plasmid, induction of *tetA*(A) from *tet*(A)$^{\Delta tetR}$ by TGC would still lead to *tetA*(A) expression levels that are lower than those observed when *tet*(A)$^{wt}$ is present at one copy. Induction of *tetA*(A) from *tet*(A)$^{wt}$ present at one copy has no significant impact on MIC to TGC (S6 Table), thus explaining why isolates carrying *tet*(A)$^{\Delta tetR}$ fail to evolve clinically relevant TGC resistance at a high frequency. The deletion in TetR(A) removes 8 amino acids, from the 8 α-helix and from an unstructured region between α-helixes 8 and 9. This region is close to the dimerization face and the binding site of tetracycline antibiotics (PDBe reference number 5mru for TetR(A) with TC (S10 Fig) and 4abz for TetR(D) with TGC). Our findings agree with previous studies showing that mutations in this interhelix region decrease induction by tetracycline in TetR(B) (52.97% amino acid sequence homology with TetR(A) [38–41]). Our data indicates that the deletion in TetR(A) might affect the binding of each tetracycline drug to TetR(A) differently or might disturb the downstream effects on TetR(A) activity of the binding of each tetracycline antibiotic differently.

## Why are the deletions in *tetR*(A) frequently observed in clinical isolates?

We found that several factors contribute to the high level of occurrence (>20% of *tet*(A)-carrying isolates) of the mutated *tetR*(A) allele in *E. coli*, especially in certain strain collections (e.g., UPEC and/or blood-isolate collections). One such factor is clonal spread due to the linkage of *tet*(A)$^{\Delta tetR}$ to the successful clone ST131. Indeed, in the NCBI isolates and the UPEC and blood strain collection, we observed that the probability of an isolate carrying the *tet*(A)$^{\Delta tetR}$ allele to be an ST131 clone was higher than the probability of any isolate to belong to ST131. Additionally, analysis of the NCBI isolates revealed an association of *tet*(A)$^{\Delta tetR}$ with specific plasmids in ST131 isolates. Although we did not test this hypothesis, these plasmids might be particularly successful or highly transmissive plasmids, as have been observed in ST131 [42]. The successful spreading of plasmids carrying *tet*(A)$^{\Delta tetR}$ could therefore play an indirect role in the spread of *tet*(A)$^{\Delta tetR}$ among ST131 isolates. However, we also observed that isolates carrying *tet*(A)$^{\Delta tetR}$ belonged not only to ST131 but also several other ST groups and that isolates of the ST131 group also carried the wild-type *tet*(A) allele on other specific plasmid types. Therefore, while clonal spread is evidently a significant factor contributing to the high frequency of *tet*(A)$^{\Delta tetR}$, there could still be other factors affecting the dissemination of this allele.

Using competition experiments, we observed a clear fitness advantage of the *tet*(A)$^{\Delta tetR}$ allele over *tetA*(A)$^{wt}$ at low antibiotic concentrations, where strains carrying *tet*(A)$^{\Delta tetR}$ had up to an 8% fitness advantage depending on the tetracycline drug and concentration. At high concentrations of tetracyclines, the benefit of increased TetA(A) levels and the associated increase in resistance outweighs the fitness cost of *tetA*(A) induction (Fig 5). In contrast, at low antibiotic concentrations, the benefit of lower induction conferred by the *tet*(A)$^{\Delta tetR}$ allele outweighs the benefit of increased resistance of *tet*(A)$^{wt}$. Thus, there is a trade-off where the wild-type allele is enriched at high concentrations of tetracyclines, while the mutant allele is selected in environments with low concentrations of tetracyclines.

Three similar 6-bp regions present at the site of the 24-bp deletion in *tetR*(A) (Fig 3B) could lead to intrinsically high rates of spontaneous deletions in *tetR*(A), thus increasing the occurrence of the *tet*(A)$^{\Delta tetR}$ allele. Indeed, using evolution experiments, we directly observed the evolution and positive selection of mutants carrying *tet*(A)$^{\Delta tetR}$ that appeared in populations

originally carrying the wild-type $tet$(A) allele. Likely, additional factors inherent to evolution affected our evolution experiments. For example, adaptation of *E. coli* MG1655 to MH medium has been observed and described before, with adapted mutants taking over the culture following prolonged growth in MH broth [43]. Additionally, the parental strain used in our evolution experiments expressed a fluorescent protein which got lost during the evolution experiment, likely due to a cost of expressing this marker. Finally, during growth at sub-MIC concentrations of tetracyclines, other mutations than the 24-bp deletion in $tetR$(A) are expected to appear and increase fitness by decreasing the cost of $tetA$(A) expression, such as mutations decreasing the intracellular concentration of minocycline [44]. All those mutations —media adaptation, loss of the fluorescence marker, and mutations decreasing $tetA$(A) induction—could lead to more fit populations still carrying the wild-type $tet$(A), which would affect the competition dynamics and prevent mutants with spontaneous 24-bp deletions in $tetR$(A) from taking over the population as rapidly as might have naively been expected from the results of competition experiments. However, our results indicate that the 24-bp deletion in $tetR$(A) was frequent enough to be detected in numerous independent cultures and that the spontaneous mutants were enriched to a large proportion of the population, especially in cultures performed in presence of low amounts of MI. Interestingly, we also observed, albeit to a lesser degree, enrichment of the 24-bp deletion in $tetR$(A) in absence of antibiotic, in contrast to the findings of our competition experiment, indicating a more complex dynamic of selection during our evolution experiments. Furthermore, despite our evolution experiments revealing that spontaneous 24-bp deletions in $tetR$(A) were frequent enough to appear in 14 out of 15 evolution cultures, the striking similarities between the local genetic contexts around $tet$(A)$^{\Delta tetR}$, as well as the differences between genetic contexts around $tet$(A)$^{wt}$ and $tet$(A)$^{\Delta tetR}$ (S7 Fig), indicate that the observed high frequency of $tet$(A)$^{\Delta tetR}$ among collections of *E. coli* isolates results from the spread of a limited number of mutated allele(s) rather than from the spread of numerous $tet$(A)$^{\Delta tetR}$ alleles that independently evolved by mutation of $tet$(A)$^{wt}$, which would have led to more similar genetic contexts for both alleles. Nonetheless, our data shows that the high association of $tet$(A)$^{\Delta tetR}$ with ST131, the horizontal spread of $tet$(A)$^{\Delta tetR}$ carried on conjugative plasmids, and the positive selection of $tet$(A)$^{\Delta tetR}$ over $tet$(A)$^{wt}$ in presence of low amounts of tetracyclines are all together likely factors explaining the high occurrence of $tet$(A)$^{\Delta tetR}$ alleles among *E. coli* isolates.

Numerous environments exist that might select for either $tet$(A)$^{wt}$ or $tet$(A)$^{\Delta tetR}$. For example, at the therapeutic levels of DC in serum (3 to 5 mg/L), selection of $tet$(A)$^{wt}$ over $tet$(A)$^{\Delta tetR}$ is expected to occur (S6 Table and Fig 5) [45]. However, therapeutic use of tetracyclines in humans and animals may also lead to selection for $tet$(A)$^{\Delta tetR}$, as poor drug pharmacokinetics, low compliance, or use of low-activity drugs can cause lower overall drug concentrations [46,47]. Furthermore, the wide use of tetracyclines in agriculture for growth promotion has led to contamination of water and soil, resulting in additional environments with drug levels in the low μg/L to low mg/L range where positive selection for $tet$(A)$^{\Delta tetR}$ over $tet$(A)$^{wt}$ is expected to occur (Fig 5) [48–50].

## A screen to predict TGC resistance evolvability

We showed that presence of $tet$(A)$^{wt}$ correlates with a high frequency of tigecycline resistance development and that isolates carrying $tet$(A)$^{wt}$ could develop resistance above clinical resistance levels. Therefore, we developed a multiplexed PCR screen that rapidly and correctly detects the $tet$(A)$^{wt}$ allele and could inform clinicians on the potential of an isolate to develop TGC resistance through amplifications of $tet$(A)$^{wt}$. Approximately 89.5% of the strains carrying the $tet$(A)$^{wt}$ allele that we tested developed TGC resistance in vitro at a high frequency

(Fig 1B and S2 Table), indicating a good predictive power and relevance of our screen. However, further experiments to confirm that $tet(A)^{wt}$ amplifications occur in animal infection models and/or patients and lead to antimicrobial treatment failure are necessary to validate the clinical usefulness of our PCR screen. Reports have shown that using a higher TGC loading dose can improve treatment, and high-dose treatment has been considered for serious clinical cases [51,52]. Potentially, this high-dose approach could be used whenever $tet(A)^{wt}$ is detected to lower the risk of developing TGC resistance. Further experiments are needed to examine the benefit of such treatment.

## Conclusion

One major finding in this work is that low-level (sub-MIC) antibiotic exposure can enrich for a mutation that prevents subsequent evolution of high-level resistance to the same antibiotic family. This is in contrast to what is generally observed during resistance evolution at sub-MIC levels where an initial resistance mutation facilitates the stepwise accumulation of mutations resulting in high-level resistance [29,53,54]. It is notable that the mutation examined here, despite being selected for by an antibiotic, does not confer an increase in resistance. Instead, this mutation is beneficial over the non-mutated gene at low antibiotic concentrations because it prevents costly induction of a resistance mechanism when the cost of expression exceeds the benefit of increased resistance. Indeed, selection of a mutation conferring a lower level of resistance combined with a lower fitness cost has previously been observed in TEM-type ESBLs [55]. Inducible antibiotic resistance mechanisms are common and observed for several antibiotic classes, including ß-lactams, macrolides, tetracyclines, aminoglycosides, and glycopeptides, as well as for heavy metals [56–59]. There could be similar sub-MIC selection for mutations in repressor proteins in other systems that reduce the cost by lowering induction. These might affect resistance at higher levels of the drug or prevent the development of resistance by gene amplification, a common mechanism of resistance [3]. Further work is needed to examine the generality of this phenomenon.

## Material and methods

### Isolates, determinants, and growth conditions

*E. coli* isolates used in this study are described in S10 Table. The wild-type allele of *tet*(A) corresponds to the sequence described in [60] and is characterised by the presence of a full-length *tetR*(A). Unless otherwise specified, cultures were grown in Mueller–Hinton broth (MHB) or MH agar (MHA) (Becton, Dickinson and Company) at 37°C, with 190 rpm shaking for broth cultures. All strains used were stored at −80°C in 10% dimethyl sulfoxide (DMSO). Antibiotics were purchased from Sigma-Aldrich (now Merck) with the exception of TGC, which was purchased through Apoteket as the Tygacil drug (Wyeth), and omadacycline, which was purchased through MedChemExpress. All work and incubations with tetracycline antibiotics were performed protected from light as much as possible to prevent antibiotic degradation.

### Simplified fluctuation assay

Duplicate overnight cultures in MHB were diluted 1:1,000 in PBS. For each diluted culture, 4 growths in 1 ml fresh media were inoculated with 1 μl of diluted cultures (about $4 \times 10^3$ bacteria), for a total of 8 biological replicates per strain. Growths were incubated for 20 hours (±30 minutes) at 37°C under vigorous shaking. Various culture volumes and dilutions were then plated on MHA plates containing tigecycline at 0.25, 0.5, 1, and/or 2 mg/L. Plates were incubated 48 hours (±2 hours) at 37°C before counting resistant mutant colonies. Total colony-

forming unit (CFU) was also determined on MHA plates without TGC. Mutant frequencies (number of mutants over the total number of cells plated) were calculated and averaged for the 8 replicates. If no mutants were observed, the maximum potential frequency was calculated by determining the frequency if one mutant had been observed in the replicate with the highest CFU plated.

## MIC determinations

Minimum inhibitory concentrations (MICs) were determined as described in Knopp and colleagues [61] using Etests (BioMérieux) and MHA plates. Etests were read according to instructions from BioMérieux and MIC determined as the median of biological triplicates, except for MICs of TGC-resistant mutants. For the TGC-resistant mutants selected, the MIC of 2 to 8 mutants was determined as follows for each selection step. The mutants were grown with (presence of TGC at the same concentration as the mutant were selected on) and without selective pressure to maintain potential unstable phenotypes, and MIC was determined using a single Etest for each growth condition. MICs were then presented as a range (no significant difference in MIC with or without selection pressure was observed, see example in S11A Fig). Broth microdilution is the recommended method of MIC determination rather than Etests, according to EUCAST [62]. To confirm that the accuracy of Etests was sufficient for our purpose, Etest and broth microdilution MICs were compared (S3 Method). Etests were deemed useable throughout this study to determine MICs (S11A Fig). MIC determinations for omadacycline, oxytetracycline, and chlortetracycline were determined using broth microdilution, as described in S3 Method.

## Frequency analysis and sequence comparison

The occurrence of *tet*(A) alleles was analysed in silico in a collection of whole-genome sequenced UPEC and blood isolates [31]. Isolates in this collection were considered genetically identical (duplicates) if the pairwise differences between the 2 isolates (data in Supplementary File S3 of [31]) was less than 15 and the isolates were of the same origin according to their position in the phylogenetic tree in Fig 1B of the same work. Duplicates were analysed in this study as a single sample. Isolates carrying *tet*(A) were identified using the whole-genome sequence data (contigs) of each isolate. Hits with sequences of high enough quality were used to determine the frequencies of the different *tet*(A) alleles. Occurrence and frequency of *tet*(A) alleles was also analysed in an in-house collection of bacteraemia *E. coli* isolates by PCR using primers described in S11 Table.

To determine the occurrence of the *tet*(A) alleles, *tet*(A) was found in *n* = 210 complete whole-genome sequenced *E. coli* isolates and plasmids labelled as "complete" in the NCBI database using BLAST and *tetA*(A). This corresponded to all *E. coli* complete genomes and plasmids labelled as "complete" with *tet*(A) that had been deposited in NCBI at the time of the search. Ten isolates where sequences were determined to be duplicates or sequence quality was deemed insufficient were removed. The *tet*(A) loci were aligned including 1,000 base pairs up- and downstream from *tet*(A) using the CLC Main Workbench software, and the frequency of the different *tet*(A) alleles determined. Point mutations were occasionally observed in *tet*(A), but not described in Figs 3 or S5, S6 or S7. MLST typing was performed using MLST 2.0 [63] (*E. coli* scheme 1), and plasmid typing was performed using Plasmid MLST [64] for sequences from NCBI where a complete genome was available (*n* = 95). The extracted MLST allele sequences from MLST 2.0 were joined and aligned using the CLC Genomics Workbench software (gap open cost = 10; gap extension cost = 1; end gap cost = as any other; alignment mode = very accurate, QIAGEN Bioinformatics), with isolates described in S1 Table included

for comparison. From the alignment, a phylogenetic tree was created using the CLC Genomics Workbench software (QIAGEN Bioinformatics, Maximum likelihood phylogeny 1.3, Neighbor Joining method, Jukes Cantor model) before editing using iTOL v6 [65].

To compare genetic contexts of *tet*(A) determinants, we used *tet*(A)-carrying isolates from NCBI where the ST had been determined. Only *tet*(A)$^{wt}$ and *tet*(A)$^{\Delta tetR}$ (with the 24-bp deletion in *tetR*(A) and no other mutations) were used (*n* = 65). For each *tet*(A) determinant, we selected a sequence spanning from 7 kb upstream to 7 kb downstream of *tet*(A). To only compare the genetic context around *tet*(A) and *tet*(A)$^{\Delta tetR}$, we first corrected the 24-bp deletion in *tetR*(A) of *tet*(A)$^{\Delta tetR}$ alleles and then aligned the sequences using CLC Genomics Workbench (gap open cost = 4; gap extension cost = 1; end gap cost = free; alignment mode = very accurate). The alignment was used to generate a maximum likelihood phylogeny tree using CLC Genomics Workbench (Neighbor Joining method, Jukes Cantor model) before editing the tree using iTOL v6. Within the sequences analysed, large identical repeats present in the same orientation and surrounding *tet*(A) were detected by aligning the 7 kb upstream and 7 kb downstream regions of *tet*(A) using CLC Genomics Workbench; 100% identical repeats were then systematically searched around all other *tet*(A) determinants analysed.

## PCR screening

*E. coli* isolates that had not previously been sequenced were screened for presence of *tet*(A), and, if applicable, the nature of the *tet*(A) allele. Template for PCR reactions was prepared by pelleting 30 to 50 μl of overnight cultures, resuspending cells in an equal volume of molecular biology grade water (Sigma-Aldrich), and heating for 10 minutes at 98˚C. PCR screening was performed using DreamTaq Green PCR Master Mix (Thermo Fisher Scientific), according to manufacturer's protocol with 2 μl DNA template per reaction. Primers (Sigma-Aldrich) are described in S11 Table. For the multiplexed screen for *tet*(A) presence and *tetR*(A) allele, the protocol used was as follows (per reaction): 12.5 μl MasterMix, 1 μl of each primer (tetAscreen_P2, tetAscreen_P4, tetRdelscreen_short_P1, tetRdelscreen_short_P2, all at 10 mM concentration; see S11 Table), 6.5 μl water, and 2 μl DNA template. Unless specified otherwise, the PCR program was as follows: 1 minute at 95˚C; 34 cycles of 94˚C for 30 seconds, 60˚C for 30 seconds, and 72˚C for 1 minute; 7 minutes at 72˚C followed by cooling to 4˚C. PCR machines used were C1000 Touch Thermal Cycler (Bio-Rad) and GeneAmp PCR System 9700 (Applied Biosystems), interchangeably. PCR products were analysed by electrophoresis in 1% agarose gels, except the multiplex PCR products which were analysed using 2% agarose gels.

## Conjugation of clinical resistance plasmids

Plasmids were extracted using Omega Bio-tek's E.Z.N.A. Midi plasmid extraction kit, according to manufacturer's protocol. Extracted plasmids were electroporated (Bio-Rad Gene Pulser Xcell, 1.8 kW, 25 μF, 200 Ω, 1 mm cuvettes) into restriction-negative, modification-positive Neb5-α electrocompetent *E. coli* (New England Biolabs). Transformants were selected with trimethoprim (15 mg/L) or streptomycin (50 mg/L, only for p083_CORR). Once presence of plasmid was confirmed by PCR (S11 Table for primers), presence of the complete plasmid and absence of other plasmids from the parental isolate was confirmed as follows. Plasmids were extracted, digested by appropriate restriction enzymes (Thermo Fisher Scientific), and subjected to electrophoresis in a 1% agarose gel to confirm presence of the plasmid of interest alone. Thereafter, the plasmid was electroporated into a diaminopimelic acid deficient (DAP⁻, grown with 20 mg/L DAP, from Sigma-Aldrich), chloramphenicol-resistant MG1655 isolate carrying sYFP2 (DA38929 [66]). Conjugations into MG1655 were performed as described in Rajer and Sandegren [66] at a 1:1 ratio. Fractions of the resuspended cells were plated on

MHA with trimethoprim (15 mg/L) or streptomycin (only for p083_CORR, 50 mg/L) and incubated overnight at 37°C. Potential successful conjugants were checked for absence of fluorescence under UV light (Visi-Blue Transilluminator, UVP) and were clean-streaked before confirming presence of the plasmid by PCR.

## Sequencing and sequence analysis

DNA for WGS was prepared using MasterPure Complete DNA and RNA Purification Kit (Epicentre or Lucigen) and 500 µl of overnight cultures. Final DNA concentration and quality was determined using Nanodrop 1000 (Thermo Fisher Scientific) and Qubit 2.0 fluorometer (Invitrogen). WGS was performed using Illumina MiSeq (in-house, described in [3]) and results were analysed with the CLC Genomics Workbench software (QIAGEN Bioinformatics). Nucleotide variations, insertions, deletions, and changes in coverage (i.e., caused by spontaneous duplications and amplifications) were determined by mapping reads from the mutants to the reference genomes of the parental strains (available on NCBI). Variations were identified using the basic variant detection tool in the CLC Genomics Workbench, with a ploidy of 1, minimal frequency of 75%, and minimal coverage of 10 reads. Insertions and deletions were screened for using the InDel tool, and amplifications were identified by manually screening for changes in coverage for the mapped reads using the CLC Genomics Workbench. Amplifications and plasmid copy numbers were determined by comparing sequencing depths (determined using CLC Genomics Workbench). For amplification copy number determination, sequencing depth over the amplified unit was normalised to the sequencing depth over the chromosome (for amplifications on the chromosome) or over the non-amplified part of the plasmid (for amplifications on plasmids). For plasmid copy number determination, sequencing depth of the plasmid (excluding potential tandemly amplified regions) was normalised to that of the chromosome.

Local sequencing was performed using Mix2Seq Kits (Eurofins Genomics), according to manufacturer's instructions. Templates for sequencing (PCR products) were purified with SureClean (Bioline), according to the manufacturer. Sequencing results were analysed with the CLC Main Workbench software.

## Strain constructions

Lambda red recombination or P1 transduction was used to construct strains [67,68]. Inserts for lambda red recombination were produced using Phusion High-Fidelity DNA polymerase (Thermo Fisher Scientific), according to manufacturer's instructions (see S11 Table for primers and single-strand oligos used for recombination). In general, lambda red recombination was performed as described in Knopp and colleagues [61] in strains carrying pSIM10 (expressing lambda red recombinase and carrying hygromycin resistance; from Court lab [69]) with the following adaptations. Approximately 50 to 100 mg/L hygromycin was added for plasmid maintenance and recovery was overnight at 30°C. Apparent successful recombinants were clean-streaked before confirmation of the correct mutation by PCR screening and local sequencing. Correct strains were grown at 37°C overnight and subsequently screened for loss of the pSIM10 plasmid.

Lambda red recombination was used to reconstruct the different $tet$(A) alleles, starting from MG1655 carrying pEC958 (with a $tet$(A)$^{wt}$) and the lambda red expression plasmid pSIM10 (DA55746). To replace the $tet$(A)$^{wt}$ by the $tet$(A)$^{\Delta tetR}$ allele (strain DA59616), an $amilCP$-$\Delta catsacB$ cassette (template: DA35728 [70]) was inserted in $tetR$(A) on pEC958 of DA55746. Transformants were selected on 12.5 mg/L chloramphenicol and screened for purple colour (presence of $amilCP$). Single-strand recombination was then performed with the

oligo tetRdel_sstrans (2 μl at 10 nM), with selection for white colonies (loss of *amilCP*) on LA sucrose plates (1% tryptone [Sigma], 0.5% yeast extract [Sigma], 1.5% agar [Sigma], 5% sucrose [Sigma]; loss of *sacB*) supplemented with 15 mg/L trimethoprim and 12.5 mg/L tetracycline to maintain selection for pEC958. Presence of pEC958 in the constructed strain was confirmed by PCR and *tet*(A) was sequenced to confirm the desired allele before growing at 37°C to lose pSIM10.

Lambda red recombination was also used to transfer *tet*(A) alleles to the chromosome of MG1655 to replace the cryptic *bgl* operon. *tet*(A)$^{wt}$ from DA52262 or *tet*(A)$^{\Delta tetR}$ from DA59616 was amplified in 2 overlapping inserts using primers tet(A)_tetRregion_P1 and tet(A)_tetRregion_P2 for the first insert and tet(A)_tetAregion_P1 and tet(A)_tetAregion_P1 for the second insert. In *E. coli* MG1655 carrying pSIM10, lambda red recombination was performed using 3 μl of each insert and transformants were selected on MHA with 12.5 mg/L tetracycline. Recombinants were confirmed by local sequencing.

Lambda red recombination was used to create translational fusions of the chromosomal *lacZ* with *tetA*(A) from different *tet*(A) alleles in MG1655. Inserts carrying *tetR*(A) and its promoter, the *tetA*(A) promoter, and the first 108 base pairs of *tetA*(A) were amplified using primers lacZfusins_P1 and lacZfusinsnew_P2 and DA44554 (for *tet*(A)$^{wt}$) or DA33135 (for *tet*(A)$^{\Delta tetR}$) as template. DA60952 (MG1655 with *lacI* and the first 24 bp of *lacZ* replaced by *amilCP*-A*catsacB*) was used to replace by lambda red the *amilCP*-A*catsacB* cassette with the *tet*(A) inserts (3 μl for each), resulting in wild-type *tetR*(A)-*tetA*(A)::*lacZ* and 24-bp-deletion-carrying *tetR*(A)-*tetA*(A)::*lacZ* fusions in DA66914 and DA66917, respectively. Transformants were counter-selected for loss of the *amilCP*-A*catsacB* cassette on sucrose plates. Recombinants were confirmed by local sequencing before growing at 37°C to lose pSIM5-tet.

For P1 transduction, lysates of donor strains were prepared as follows: 2 cultures of 4 ml P1 LB medium (LB broth (Sigma-Aldrich) supplemented with 20 mM MgCl$_2$, 5 mM CaCl$_2$, and 0.2% glucose) were inoculated with 40 μl overnight culture of the donor strain before incubation at 37°C for 1 hour (to early logarithmic phase) followed by the addition of 100 μl P1 phage lysate (prepared from wild-type MG1655, DA5438) and continued growth until lysis was complete. 200 μl chloroform was then added before vortexing for 10 seconds and centrifugation at 17,000 g for 1 minute. Supernatant was transferred to a new tube, 200 μl chloroform was added before vortexing for 10 seconds and centrifugation at 17,000 g for 1 minute. Prepared lysates were stored at 4°C. To perform P1 transduction into recipient strains, 400 μl overnight growths in P1 LB medium of recipient strains were incubated with 75 μl donor lysate at 37°C for 30 minutes without shaking. Cell-free and lysate-free controls were performed in parallel to control for contamination. After incubation, cells were pelleted at 3,500 g for 5 minutes, resuspended in 100 μl LB broth supplemented with 100 mM Na-citrate, and plated on LB agar (Sigma-Aldrich) supplemented with the appropriate antibiotic. The transductants were then clean-streaked before confirmation screening by PCR.

## qPCR and RT-qPCR

DNA templates were prepared as described above (see PCR screening). mRNA extractions were performed with biological triplicates according to Roemhild and colleagues [71]. The concentration of each tetracycline antibiotic used, when applicable, was 1/10th the MIC of the *tet*(A)$^{wt}$-carrying strain DA52262. qPCR was performed according to Roemhild and colleagues [71] using 2.5 μl template (of a total volume of 10 μl per reaction) for each of 2 technical replicates, an annealing temperature of 60°C, and 40 cycles. For *tet*(A) copy number determination, the control genes used were *cysG* (chromosomal) and *pemK* (plasmid pEC958), and *tet*(A)

copy numbers were normalised to the chromosomes. For mRNA expression analysis, the control genes used were *cysG*, *hcaT*, and *idnT* [72]. Oligo sequences can be found in S11 Table.

## ß-galactosidase assays

Overnight growths of MG1655 *tetR*(A)-*tetA*(A)::*lacZ* constructs with either wild-type *tetR*(A) or *tetR*(A) with the 24-bp deletion (DA66914 and DA66917, respectively) were diluted 1:100 in 10 ml MHB supplemented or not with antibiotic and grown at 37˚C under vigorous shaking to an $OD_{600}$ of 0.5 to 0.7 (approximately 2 hours). Cultures were chilled on ice for at least 10 minutes before measuring $OD_{600}$. Enzyme activity analysis was performed according to Kacar and colleagues [73] with 1 ml of each culture and the following alterations. The lysed samples were diluted 1:5 in Z-buffer (60 mM $Na_2HPO_4 \cdot 7 H_2O$, 40 mM $NaH_2PO_4 \cdot H_2O$, 10 mM KCl, 1 mM $MgSO_4 \cdot 7H_2O$, 50 mM β-mercaptoethanol (pH 7)) before transfer to Honeycomb plate wells (produced for Bioscreen C by Oy Growth Curves AB) (4 technical replicates per original culture as well as 4 negative controls with Z-buffer and ONPG only). Six biological replicates were performed each for the wild-type *tetR*(A) and *tetR*(A) with the 24-bp deletion at each antibiotic concentration tested. The plates were incubated in the Bioscreen apparatus at 28˚C with shaking before each measurement. The following equation was used to calculate the ß-galactosidase activity: $[(OD_{420}-OD_{540})/OD_{600}]-(OD_{420 \text{ control}}-OD_{540 \text{ control}})$ = relative breakdown of ONPG. The results were plotted against time and the slope was determined over the times 60 to 120 minutes after start of incubation, providing the ß-galactosidase activity. Enzyme activities following TC and TGC induction were compared at 2 hours and 6 hours of growth for both alleles of *tetR*(A) to observe if there was a delay in diffusion of the bulkier TGC antibiotic into the cell. No relevant difference in diffusion rate was observed (S11B Fig) and TC penetration in wild-type cells without tetracycline-specific efflux reaches a plateau after ±20 minutes, indicating that induction for 2 hours was already optimal [33].

## Competitions

MG1655 *bgl*::*tet*(A)$^{wt}$ (D60962) and MG1655 *bgl*::*tet*(A)$^{\Delta tetR}$ (DA60964) were fluorescently labelled with both sYFP2 and mScarlett in the *ΔIS150* region (replacing MD-39 [74], a neutral deletion as described by Pósfai and colleagues [75] by P1 transduction from donor strains DA55290 and DA55286, respectively). DA55290 and DA55286 were used as isogenic strains lacking *tet*(A) in competitions. Constructed competition strains are presented in S12 Table.

Overnight cultures of competition isolates were mixed 1:1 before inoculating 180 μl MHB (with or without antibiotic) with 1.4 μl mixed culture. Antibiotic concentrations tested were fractions of the initially determined MIC of MG1655 *bgl*::*tet*(A)$^{wt}$ (DA60962; MIC: 32 mg/L TC, 0.125 mg/L TGC, 12 mg/L DO, 3 mg/L MI, 96 mg/L OT, and 1,536 mg/L CT). Competitions were performed with 12 to 13 biological replicates for each fluorescent marker combination, and competitions with strains where the fluorescence markers had been swapped were also tested for each antibiotic concentration. This resulted in a total of 18 to 25 biological replicates for each antibiotic concentration and competition. Cultures were incubated over 3 days with 1.4 μl of the previous culture inoculated in 180 μl fresh media every 24 (±1 hour) for 7 generations of growth each day. At every 24-hour period, 1 μl culture was transferred to 200 μl sterile PBS and incubated at room temperature for at least 30 minutes before determining fluorescence in individual cells by flow cytometry (MACS instrument, Miltenyi Biotec). Replicates showing signs of contamination (identified visually and by the combined gating of both fluorescence markers comprising less than 90% of the total cells counted) were discounted. For high antibiotic concentrations where growth of both strains was visibly impaired (1/2 and 1/4 × MIC of strain carrying the *bgl*::*tet*(A)$^{wt}$), replicates where one isolate had clearly

developed resistance improving growth were discounted. Selection coefficients and the average thereof were determined by CompDAta v4.4. The fitness cost for the fluorescent markers in each background was removed from the selection coefficients. For this, cost of the fluorescent markers was determined by competing at 1/32 times MIC for each antibiotic (average of 9 replicates) isogenic isolates expressing sYFP2 or mScarlett (DA55286 and DA55290 for isolates lacking *tet*(A), DA60990 and DA62534 for wild-type *tet*(A), and DA60999 and DA62537 for *tet*(A)$^{\Delta tetR}$).

### Evolution and amplicon sequencing analysis

Twenty lineages of DA60990 with sYFP2 and *tet*(A)$^{wt}$ were passaged twice a day in MHB with and without 0.0469 mg/L minocycline for 14 days, with 20 μl culture inoculated in 2 ml fresh media at each passage (about 7 generations of growth per passage, for a total of 196 generations). This resulted in 20 lineages grown in two conditions and a total of 40 growths at the end point of 28 passages. The 40 growths were screened for detectable proportions of *tet*(A)$^{\Delta tetR}$ by PCR, as described above.

To more accurately determine the proportion of *tet*(A)$^{\Delta tetR}$ in a subset of linages, amplicon sequencing of the region using as template *tet*(A)$^{wt}$ was performed as described in [76] with the following exceptions: DNA was prepared as described in main text under "PCR screening" for WGS. Primers were barcoded and are described in S11 Table. For each sample, two amplicons were amplified and sequenced (technical replicates). Analysis of the amplicon sequencing was performed using the CLC Genomics Workbench software (Qiagen). Reads were mapped to 90 bp upstream and downstream of the deletion site in *tet*(A)$^{wt}$ and *tet*(A)$^{\Delta tetR}$ variant separately with the following settings: no masking, match score = 3, mismatch cost = 3, affine gap cost mode, insertion open cost = 8, insertion extend cost = 3, deletion open cost = 8, deletion extend cost = 3, length fraction = 0.3, similarity fraction = 0.98. For each amplicon sequencing, the frequency of bacteria with *tet*(A)$^{\Delta tetR}$ was calculated by dividing the sequencing coverage for the *tet*(A)$^{\Delta tetR}$ reference by the total sequencing coverage for both *tet*(A) variants.

### Statistical analysis

All statistical tests were performed using R (3.5.0) and/or GraphPad Prism (8.3.1). All statistical tests were performed as 2-sided tests.

### Supporting information

**S1 Table. Phylogeny of isolates used in initial screen for evolution of TGC resistance.**
(XLSX)

**S2 Table. Frequency and phenotype of TGC-resistant mutants selected at TGC 0.5 mg/L.**
(XLSX)

**S3 Table. Frequency and phenotype of TGC-resistant mutants selected at TGC 1 and/or 2 mg/L.**
(XLSX)

**S4 Table. Lineages, MICs of mutants, and mutations found following selection of mutants at TGC 1 or 2 mg/L and following stepwise selection of mutants in presence of increasing TGC concentrations.**
(XLSX)

**S5 Table. MICs of mutants and mutations found following selection of resistant mutants at TGC 0.5 mg/L.**
(XLSX)

**S6 Table. MICs of a subset of isolates.**
(XLSX)

**S7 Table. Effect of *tet*(A) alleles on TGC resistance development.**
(XLSX)

**S8 Table. Characterisation of *tet*(A)-carrying isolates from NCBI.**
(XLSX)

**S9 Table. Evolution of *tet*(A) experiments.**
(XLSX)

**S10 Table. Strains used in this study.**
(XLSX)

**S11 Table. Primers used.**
(XLSX)

**S12 Table. MICs of strains carrying constructs used in competition studies.**
(XLSX)

**S1 Fig. Stepwise selection of TGC mutants on progressively increasing TGC concentrations.**
(PDF)

**S2 Fig. Effect of *tet*(A) overexpression and copy number increase.**
(PDF)

**S3 Fig. Effect of *tet*(A)$^{wt}$ amplifications on TGC MIC in a clinical *Klebsiella pneumoniae* and omadacycline MICs in *E. coli*.**
(PDF)

**S4 Fig. Effect of *tet*(A)$^{\Delta tetR}$ amplification on TGC and omadacycline MIC.**
(PDF)

**S5 Fig. Schematic of all observed *tet*(A) alleles.**
(PDF)

**S6 Fig. Phylogeny of *tet*(A)-carrying isolates from NCBI.**
(PDF)

**S7 Fig. Phylogeny of genetic context surrounding *tet*(A) in isolates from NCBI.**
(PDF)

**S8 Fig. PCR screen to detect presence of *tet*(A) and differentiate the *tet*(A) alleles.**
(PDF)

**S9 Fig. Competition of a strain carrying *tet*(A)$^{\Delta tetR}$ vs. an isogenic strain without any *tet*(A) allele at different antibiotic concentrations.**
(PDF)

**S10 Fig. Protein structure and site of 8 amino acid deletion in TetR(A).**
(PDF)

**S11 Fig. Method control experiments.**
(PDF)

**S1 Results. Association of *tetR*(A) mutation with ST131 group.**
(PDF)

**S2 Results. Optimization of multiplex PCR screen.**
(PDF)

**S1 Method. Stepwise selection for increased TGC resistance.**
(PDF)

**S2 Method. TGC MIC and fitness cost with overexpression of *tetA*(A).**
(PDF)

**S3 Method. Comparison of Etests and broth microdilution MIC determination.**
(PDF)

**S1 Data. Additional data for figures and tables.**
(XLSX)

## Acknowledgments

We would like to thank Pr. Mathew Upton (University of Plymouth, United Kingdom) for strain EC958; Dr. Brian Coombes (McMaster University, Ontario, Canada) for strain NRG857c; Dr. Fernando Ruiz-Perez (University of Virginia, Virginia, United States of America) for strain 042; Pr. Ulrich Dobrindt (University of Würzburg, Würzburg, Germany) for strain 536; Dr. Nicolas Barnich (UMR Inserm/University of Auvergne U1071, USC INRA 2018, Clermont-Ferrand, France) for strain LF82; Pr. Laurence Van Melderen (Free University of Brussels, Brussels, Belgium) for strain PMV01; Dr. Swaine L. Chen (Genome Institute of Singapore, Singapore) for strain CI5; Dr. Wolfram Petzl (Ludwig Maximilans University, Munich, Germany) for strain 1303; Pr. Erik Denamur (UMR 1137 INSERM, University of Paris and University Sorbonne Paris North, Paris, France) for strains iAi1, iAi39, UMN026, ED1A, and 55989; Dr. Valeria Michelacci (Italian National Institute of Health, Rome, Italy) for strain CFSAN029787, Pr. Catharina Svanborg and Inès Ambite (Lund University, Lund, Sweden) for strain ABU 83972; the International Center for Genetic Engineering and Biotechnology, Trieste, Italy for strain Nissle1917; Dr. Linus Sandegren (Uppsala University, Uppsala, Sweden) for strains 2/0/p2_0.1, 4/0/p4_0.1, 4/1-1/p4/1-1.1, 4/1-2/p4/1-2.1, and 4/4/p4/4.1; and Dr. Surbhi Malhotra (University of Antwerp, Antwerp, Belgium)—SATURN network contract FP7-HEALTH-2009-SINGLE STAGE-no.241796 for strain ST540. We would also like to thank Dr. Stephen J. Salipante (University of Washington, Washington, USA) for contributing 15 strains from his uropathogenic (UPEC) and blood isolates collection. Finally, we thank Dr. Nikolaos Fatsis-Kavalopoulos and Dr. Lionel Guy for their recommendations regarding statistical analysis and Dr. Arianne Babina for her recommendations regarding language.

## Author Contributions

**Conceptualization:** Jennifer Jagdmann, Dan I. Andersson, Hervé Nicoloff.

**Formal analysis:** Jennifer Jagdmann.

**Funding acquisition:** Dan I. Andersson.

**Investigation:** Jennifer Jagdmann.

**Methodology:** Jennifer Jagdmann, Hervé Nicoloff.

**Supervision:** Dan I. Andersson, Hervé Nicoloff.

**Visualization:** Jennifer Jagdmann.

**Writing – original draft:** Jennifer Jagdmann, Dan I. Andersson, Hervé Nicoloff.

**Writing – review & editing:** Jennifer Jagdmann, Dan I. Andersson, Hervé Nicoloff.

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
