## [Editor Report · Decision Letter 0]

10 Mar 2022

Dear Dr Nicoloff, 

Thank you for submitting your manuscript entitled "Antibiotic selection of a mutation preventing the evolution of high-level resistance" for consideration as a Research Article by PLOS Biology.

Your manuscript has now been evaluated by the PLOS Biology editorial staff, as well as by an academic editor with relevant expertise, and I'm writing to let you know that we would like to send your submission out for external peer review.

Once your full submission is complete, your paper will undergo a series of checks in preparation for peer review. Once your manuscript has passed the checks it will be sent out for review. To provide the metadata for your submission, please Login to Editorial Manager (https://www.editorialmanager.com/pbiology) within two working days, i.e. by Mar 14 2022 11:59PM.

If your manuscript has been previously reviewed at another journal, PLOS Biology is willing to work with those reviews in order to avoid re-starting the process. Submission of the previous reviews is entirely optional and our ability to use them effectively will depend on the willingness of the previous journal to confirm the content of the reports and share the reviewer identities. Please note that we reserve the right to invite additional reviewers if we consider that additional/independent reviewers are needed, although we aim to avoid this as far as possible. In our experience, working with previous reviews does save time. 

If you would like to send previous reviewer reports to us, please email me at rroberts@plos.org to let me know, including the name of the previous journal and the manuscript ID the study was given, as well as attaching a point-by-point response to reviewers that details how you have or plan to address the reviewers' concerns. 

Given the disruptions resulting from the ongoing COVID-19 pandemic, please expect some delays in the editorial process. We apologise in advance for any inconvenience caused and will do our best to minimize impact as far as possible.

Kind regards,

Roli Roberts

Roland Roberts

Senior Editor

PLOS Biology

rroberts@plos.org

---

## [Decision Letter · Decision Letter 1]

20 May 2022

Dear Hervé,

Thank you for your patience while your manuscript "Antibiotic selection of a mutation preventing the evolution of high-level resistance" went through peer-review at PLOS Biology. Your manuscript has now been evaluated by the PLOS Biology editors, an Academic Editor with relevant expertise, and by three independent reviewers. Please accept my further apologies for the technical difficulties that delayed our discussion of the decision with the Academic Editor; we're seeking to rectify these issues.

You’ll see that the reviewers are all broadly positive about your study, but between them they have a number of requests that will need to be addressed before further consideration; most of these are textual in nature, but some of them will likely require additional analyses. In addition, the Academic Editor has also made a minor suggestion (see the comments at the foot of the letter) - you will be able to identify the Academic Editor by this comment, and it should be treated as optional.

In light of the reviews, which you will find at the end of this email, we are pleased to offer you the opportunity to address the comments from the reviewers in a revision that we anticipate should not take you very long. We will then assess your revised manuscript and your response to the reviewers' comments with our Academic Editor aiming to avoid further rounds of peer-review, although might need to consult with the reviewers, depending on the nature of the revisions.

**IMPORTANT - SUBMITTING YOUR REVISION**

*Resubmission Checklist*

*Published Peer Review*

*PLOS Data Policy*

*Blot and Gel Data Policy*

Sincerely,

Roli

Roland Roberts

Senior Editor

PLOS Biology

rroberts@plos.org

REVIEWERS' COMMENTS:

Reviewer #1:

Very complete and interesting work showing that, in contrast to the norm, some mutations selected in the presence of subinhibitory concentrations of antibiotics may prevent the evolution of high level resistance to the same class of antibiotics. The findings are scientifically relevant and may as well have implication for guiding treatment of relevant infectious diseases by MDR pathogens.

1. Results, subheading identification of TGC resistance mechanisms. Were basal MIC homogeneous? Any impact of basal MIC on MF or resistance development? Did tet(A) positive show higher basal MICs?

2. Lines 127-150. Was gene dosage amplification the only mechanism leading to increased expression? Correlation with transcription data?

3. Lines 183-186. Was tet(A) expression similar in tetR wt and delta evolved strains in addition to having similar gene dosages?

4. Not fully clear what is the effect of a fully truncated tetR on tetA basal or induced expression as compared with delta 24.

5. If tetR delta 24 fails to increase tetA expression in the presence of TGC should this not have an impact on susceptibility/resistance to the drug?

6. Lines 337-338. Why not TGC?

Reviewer #2:

In this manuscript by Jagdmann et al., the authors report a new mechanism of high-level tigecycline resistance through gene amplification of tet(A), that development of high-level resistance through this pathway is not available to strains that have a deletion in tetR(A), and that this deletion appears to be selected by sub-inhibitory concentrations of tetracyclines. Overall, this is an interesting and nicely done study. I have a few recommendations and suggestions. 

Line 70-71. Please clarify what 'moderate reduction' means quantitatively. Similarly, please report the clinical breakpoint for TGC and the quantitative definition of "high TGC MICs" and "high-level TGC resistance". These appear in the 'Discussion' section, but it would be helpful to have in the introduction.

Lines 71-73. Are TGC resistance mutations described in other Enterobacteriales? On a quick perusal of the literature, it looks like TGC treatment failure due to tet(A) mutations has been described in K. pneumoniae (e.g., PMID 33717043).

For the isolates with tet(A), were all the sequences of tet(A) identical? For the 53 tet(A) isolates used in the screen, could you provide a phylogeny based only on the tet(A) sequence so we could understand how similar they are? Or were all those marked as 'wt' actually identical? Similarly, are those with the 24bp del in tetR(A) otherwise genetically identical?

In the isolates with amplification of tet(A), were the copies exact, or were there mutations in the amplified tet(A)?

Lines 136-8. While overexpression can lead to increased TGC MIC, to show that this explains that the amplifications were associated with increased tet(A) dosage, do you have transcriptional data showing that there was in fact increased tet(A) expression in the clinical isolates? 

Figure S1. What does 'amplification 9.5x' refer to (at about 6 o'clock on the circle in the figure)? In what way is there a half of an amplification?

Lines 159-161. For the K. pneumoniae experiments, were the amplifications sequenced or only counted by PCR?

Lines 161-166. Please indicate in the text and/or on Fig S4 the breakpoint for omadacycline resistance.

Section about prevalence. I'm confused about this '200' isolates in the NCBI E. coli genome sequences, as my quick check suggests nearly 30,000 genome sequences. https://www.ncbi.nlm.nih.gov/genome/browse/#!/prokaryotes/167/ Where were the 200 from? Please provide a link. For collections ii and iii as sited, it'd be really helpful to see all of these on a phylogeny to understand whether the instances of tet(A) represent clonal expansion or multiple occurrences in distinct lineages. 

S1 results. It'd be helpful to see these isolates on a phylogeny. How many times has the tet(A)ΔtetR allele appeared? Is a wt allele ever regained?

Line 231-3. The potential application of this assay seems more suited to the discussion section than the results section. 

How much time does E. coli spend exposed to tetracycline? The premise is that exposure to low dose tetracyclines selects for the allele with the deletion, but the advantage of the wt allele in the absence of antibiotic suggests that it would have to be rather consistently exposed to antibiotic. Alternatively, it could be that the allele with the deletion may be linked to other factors that confer resistance or fitness advantage or be part of a successful clade. This is another instance where it would be helpful to see the phylogeny to understand how much of the expansion of the allele with the deletion might be due to clonal success vs repeated selection. 

Line 333. I think this should cite Fig S5, not S6. 

The regular appearance of the 24bp deletion in tetR(A) even in the absence of antibiotic pressure calls into question the interpretation that it's only selected for under low antibiotic pressure (lines ~478-482). 

Lines 345 and 365. Was it 192 or 196 generations? 

Lines 394-401. What evidence is there that the four isolates described here lost the amplifications as opposed to some other process—e.g., something technical like testing or recording error vs some other genetic alteration? Had the authors observed evidence of loss of amplifications in the strains they were working with, or do the authors have other evidence to suggest the amplifications are unstable?

Lines 402-4. I'm confused by claim that this is the first instance of considering genetic diversity in evolution of antibiotic resistance. See, for example, https://www.pnas.org/doi/10.1073/pnas.2016886118, and the citations therein for just some instances of preceding work demonstrating this point, at least as far as I understand the authors. In what way is this first?

Lines 424-7. Please show this as a supplementary figure and put it in the results section. An assessment of the structural impact of the mutation seems appropriate for the first description of the mutation. 

Line 433. The question, "why is this deletion prevalent in clinical isolates" raises the issue of what should be considered 'prevalent' and the issues about assessing collections as in my comment above.

Section starting line 484. The screen to predict TGC resistance through this path is a bit confusing. Is the idea that an E. coli reported as susceptible to TGC that also has wildtype tet(A) is likely to develop resistance if standard dosing is used? How different is this rate from what's observed in E. coli lacking tet(A)?

Reviewer #3:

[identifies herself as Teresa M Coque]

The work by Jagdmann, Andersson, and Nikoloff describe the mechanism of resistance to high levels of tigecycline (a last-resort antibiotic of the family of tetracyclines), consisting of the tandem amplification of the tetA(A) efflux pump at plasmid and also at chromosomal levels. It reports how this mechanism is selected at high concentrations of the antibiotic and how low concentrations in different environments led to a dead-end that prevents the evolution towards high-level tigecycline resistance. The findings described and the experimental design are novel and of potential relevance for adopting therapeutical decisions. First, it is one of the few publications approaching the evolution of antibiotic resistance taking into account the intraspecies diversity (traditional models of experimental evolution are usually performed using a single or a very few lab strains). Second, the results demonstrate how antibiotics can select mutations/"indel" changes that impair the evolution towards antibiotic resistance which could be used for screening specific changes and predicting the efficiency of antibiotic treatments. The manuscript is well organized and written although it may result difficult to follow often due to the high number of sections, and figures. Efforts should be made here. Also, some interpretations of the results should be softened.

Some comments are given for the author´s consideration.

* A revision of the subsections and figures would help to follow the paper. Different sections of the paper refer to the same figures that do not always follow an order. This would have to be carefully revised in order to facilitate the reading. 

* Lines 159-166 and Figure S2C. About Klebsiella pneumoniae isolates resistant to tigecycline and omadacycline. Isolates DA32094 and its spontaneous TGC-resistant mutant DA59363 are only mentioned in Figure S2C. The names of the strains could be also included in table S1 and in the text in order to better follow the text. Please, explain or support by appropriate reference how this "spontaneous mutation" occurs. This is a very important result that is barely exploited. 

* Lines 179-184. The tandem amplification of the tet(A) using other antibiotics such as tobramycin is really interesting and is not exploited by the authors either. Note that the MDR island of IncFII derivatives of R1 comprises Tn1721 and also other transposons encoding antibiotic resistance. Did you try other antibiotics such as beta-lactams or chloramphenicol or quinolones? Selection by antibiotics that act on different cell targets would contribute to understanding the evolution and prediction of tigecycline resistance. Some of the antibiotics mentioned are the first election drugs to treat infections caused by enterobacterales. 

* Lines 221-225 (results) and lines 439-449 (discussion). The idea of linking the success of this mechanism of tigecycline resistance to the success of widespread clones such as ST131 is tempting. However, the occurrence of these mutants might be linked to the widespread of the IncFII plasmid where Tn1721-tet(A)is located. This plasmid IncFIIpC15-1a-like and many similar derivatives are overrepresented in ST131. An MDR island located in this type of FII plasmids is a more (or equally) plausible explanation.

* Lines 227-233. This small paragraph may be a conclusion /recommendation you can extract from your results. I would strongly recommend only maintaining this point in the discussion section

* Lines 461. The hypothesis of clonal interference when plasmids are involved is difficult to maintain. Note that E. coli populations with and without plasmids often coexist in human and environmental samples and also coexist with other enterobacterales that may carry or acquire the same plasmids. Other phenomena such as bet-hedging or frequent dependent selection, more associated with the maintenance of the robustness of the populations, could also explain this

* Line 246. Please, explain "possibly" here. This result should be "black/white". If not, it should be discussed.

* Change "prevalence" by "occurrence". You do not can distinguish the cases by date-onset and the concepts of "prevalence" and "incidence" relies on the temporal parameter.

* Tables and figures. This part needs careful revision. Below, I raise the more relevant changes 

o Tables S2 and S4 are missing. At least, this reviewer could not find any excel file. If they are there, forgot the comment.

o Table S3. This table is difficult to read. If the issue is to show the phenotype diversity of the two main genotypes, the main criterium is the genotype and this must be the first column. Then, this column should show the two genotypes, first all tet(A) and afterward, all tet(A) deleted. This reorganization would help to process the information. 

o Figure 3. This figure is very confusing and should be revised. Panel D cannot be an individual panel; it must be the footnote of the figure because explains panels A-C.

COMMENT FROM THE ACADEMIC EDITOR:

I have an unusual suggestion, i.e. that the authors consider a study we recently did with TEM-1 beta-lactamase (Ruelens & de Visser 2021 Antimicrob Agents Chemother 65:e00471-21), where we demonstrated a fitness benefit of a cefotaxime-resistance mutation (R164S) at low antibiotic concentrations, whereas only at higher concentrations a mutation (G238S) causing substantially larger MIC increases becomes selectively superior. Similar to the study by Jagdmann et al., selection of R164S not only leads to modest MIC increases, but also initiates a pathway with much lower final MIC than pathways initiated with G238S. Given the emphasis Jagmann et al. put on their deviant findings, it seems appropriate to me that they consider mentioning our recent study with a similar message for beta-lactam resistance

---

## [Decision Letter · Decision Letter 2]

23 Aug 2022

Dear Hervé,

Thank you for your patience while we considered your revised manuscript "Clonal spread and antibiotic selection of a mutation preventing the evolution of high-level resistance" for publication as a Research Article at PLOS Biology. This revised version of your manuscript has been evaluated by the PLOS Biology editors, the Academic Editor, and two of the original reviewers.

Based on the reviews and our Academic Editor's assessment of your revision, we are likely to accept this manuscript for publication, provided you satisfactorily address the remaining points raised by the reviewers and Academic Editor. Please also make sure to address the following data and other policy-related requests.

IMPORTANT:

a) Please address the remaining points from reviewer #2, and the comments from the Academic Editor that appear at the foot of this email.

b) Given the comments from the reviewer and the Academic Editor, plus the need to make both the specific and conceptual advance more explicit in the Title, we suggest that you change the Title to something like "Low tigecycline levels select for a mutation that prevents the evolution of high-level resistance to this antibiotic" (but feel free to suggest your own version).

c) Many thanks for providing the underlying data. Please could you also cite its location clearly in each relevant main and supplementary Figure legend, e.g. "The data underlying this Figure may be found in S1 Data."

We expect to receive your revised manuscript within two weeks. 

*Published Peer Review History*

*Press*

Sincerely,

Roli

Roland Roberts, PhD

Senior Editor,

rroberts@plos.org,

PLOS Biology

SPECIES INDICATED IN THE ABSTRACT? 

- Please note that per journal policy, the model system/species studied should be clearly stated in the abstract of your manuscript. 

We require the original, uncropped and minimally adjusted images supporting all blot and gel results reported in an article's figures or Supporting Information files. We will require these files before a manuscript can be accepted so please prepare and upload them now. Please carefully read our guidelines for how to prepare and upload this data: https://journals.plos.org/plosbiology/s/figures#loc-blot-and-gel-reporting-requirements

DATA NOT SHOWN?

REVIEWERS' COMMENTS:

Reviewer #1:

[identified himself as Antonio Oliver]

All points raised to the previous version have been satisfactorily addressed and therefore I have no further comments for the authors consideration

Reviewer #2:

I thank the authors for their responses and work to update the manuscript. My main comment is that the title and abstract are a bit too general. The mutation they've identified is not only spread clonally, so describing this as 'clonal spread' seems misleading. Similarly, "a mutation preventing the evolution of high-level resistance" seems incomplete -- I suggest adding that this is high-level resistance specifically of tigecycline and in E. coli. In the abstract, the added language, 'we found evidence of clonal spread...' should also include mention that not all the spread was clonal, as noted in the discussion: "However, we also observed that isolates carrying tet(A)ΔtetR belonged not only to ST131 but also several other ST groups". Also in the abstract, I wonder about the addition of 'clonal spread' on line 38. As it reads, it suggests that clonal spread selected for a mutation, whereas what I think the authors mean is that clonal spread increased the frequency of representation in the databases they checked. I suggest revising this sentence.

COMMENTS FROM THE ACADEMIC EDITOR:

[lightly edited]

a) I checked the authors' response also to rev#3 and think this is satisfying. I agree with the authors that their request for further experiments with tobramycin is beyond the scope of the current work. 

b) I further agree with rev#2 that "clonal spread" should better be removed from the title and the role of clonal spread could be clarified in the abstract (although it will be clear to the reader that the data presented suggests at least a positive role of clonal spread of ST131).

c) It would help to make explicit in Fig. 1B that the presented mutant frequencies refer to a fluctuation assay, not to frequencies among (clinical) isolates

d) I spotted several typo's (e.g. a reference to Fig.2E -- which doesn't exist -- in line 376.

---

## [Editor Report · Decision Letter 3]

29 Aug 2022

Dear Hervé,

Thank you for the submission of your revised Research Article "Low levels of tetracyclines select for a mutation that prevents the evolution of high-level resistance to tigecycline" for publication in PLOS Biology. On behalf of my colleagues and the Academic Editor, Arjan de Visser, I'm pleased to say that we can in principle accept your manuscript for publication, provided you address any remaining formatting and reporting issues. These will be detailed in an email you should receive within 2-3 business days from our colleagues in the journal operations team; no action is required from you until then. Please note that we will not be able to formally accept your manuscript and schedule it for publication until you have completed any requested changes.

Sincerely,

Roli

Senior Editor

PLOS Biology

rroberts@plos.org